# Implicit Regularization via Feature Alignment

One important property of deep neural networks is their ability to generalize well on real data. Surprisingly, this is even true with very high-capacity networks *without explicit regularization* [42, 61, 29]. This seems at odds with the usual understanding of the bias-variance trade-off [24, 41, 11]. Solving this apparent paradox requires understanding the various learning biases induced by the training procedure, which can act as implicit regularizers [42, 44].

In this paper, we help clarify one such implicit regularization mechanism, by examining the evolution of the *neural tangent features* [30] learned by the network along the optimization paths. Our results can be understood from two complementary perspectives: a *geometric* perspective – the (uncentered) covariance of the tangent features defines a metric on the model function class, akin to the Fisher information metric [e.g., 2]; and a *functional* perspective – through the tangent kernel and its RKHS.

Our main observation is a dynamical alignment of the tangent features along a small number of task-relevant directions during training (Section 3), which can be interpreted as a combined *feature selection* and *compression* mechanism. The motivating intuition is that such a mechanism allows the model to adapt its capacity to the task and underpins the generalization abilities of heavily overparametrized models. Drawing upon intuitions from linear models, we motivate a new heuristic complexity measure which captures this phenomenon, and empirically show correlation with generalization (Section 4).

**Preliminaries** Let $\mathcal{F}$ be a class of functions (e.g a neural network) parametrized by $\mathbf{w} \in \mathbb{R}^P$. We restrict here to *scalar* functions $f_\mathbf{w} \colon \mathcal{X} \to \mathbb{R}$ to keep notation light.[1] We define the **tangent features** as the function gradients w.r.t the parameters, $\Phi_\mathbf{w}(\mathbf{x}) := \nabla_\mathbf{w} f_\mathbf{w}(\mathbf{x})$, which govern how small changes in parameter affect the function's outputs,

$$\delta f_\mathbf{w}(\mathbf{x}) = \langle \delta\mathbf{w}, \Phi_\mathbf{w}(\mathbf{x}) \rangle + O(\|\delta\mathbf{w}\|^2) \tag{1}$$

More formally, the (uncentered) covariance matrix $g_\mathbf{w} = \mathbb{E}_{\mathbf{x}\sim\rho}\left[\Phi_\mathbf{w}(\mathbf{x})\Phi_\mathbf{w}(\mathbf{x})^\top\right]$ acts as a **metric tensor** on $\mathcal{F}$: assuming $\mathcal{F} \subset L^2(\rho)$, this is the metric induced on $\mathcal{F}$ by pullback of the $L^2$ scalar product (see Longer Version, Appendix A). It characterizes the geometry of the function class $\mathcal{F}$. Metric (as symmetric matrices) and tangent kernels (as integral operators) share the same spectrum.

The structure of the tangent features impacts the evolution of the function during training. Given $n$ input samples, consider gradient descent updates $\delta\mathbf{w}_{\mathrm{GD}} = -\eta\nabla_\mathbf{w}L$ for some cost function $L$. The function updates $\delta f_{\mathrm{GD}}(\mathbf{x}) := \langle \delta\mathbf{w}_{\mathrm{GD}}, \Phi_\mathbf{w}(\mathbf{x}) \rangle$ in the linear approximation (9), decompose as

$$\delta f_{\mathrm{GD}}(\mathbf{x}) = \sum_{j=1}^{P} \delta f_j u_{\mathbf{w}j}(\mathbf{x}), \quad \delta f_j = -\eta\lambda_{\mathbf{w}j}(\boldsymbol{u}_{\mathbf{w}j}^\top \nabla_{\mathbf{f}_\mathbf{w}}L), \tag{2}$$

where $(u_{\mathbf{w}j})_{j=1}^{P}$ is the **eigenbasis** of the tangent kernel and $\boldsymbol{u}_{\mathbf{w}j} = [u_{\mathbf{w}j}(\mathbf{x}_1), \cdots u_{\mathbf{w}j}(\mathbf{x}_n)]^\top$. From the point of view of function space, the metric/tangent kernel eigenvalues act as a mode-specific rescaling $\eta\lambda_{\mathbf{w}j}$ of the learning rate. This is a local version of a well-known bias for linear models (see Longer Version, Appendix B.2), towards functions in the top eigenspaces of the kernel.

As a first illustration of *non-linear* effects, Fig. 3 (Longer Version) shows visualizations of eigenfunctions of the tangent kernel of a MLP trained on a simple classification task: $y(\mathbf{x}) = \pm 1$ depending on whether $\mathbf{x} \sim \mathrm{Unif}[-1, 1]^2$ is in the centered disk of radius $\sqrt{2/\pi}$. After a number of iterations, we observe (rotation invariant) modes corresponding to the class structure (e.g boundary circle) showing up in the top eigenfunctions of the learned kernel. We also note an increasing spectrum anisotropy – for example, the ratio $\lambda_{20}/\lambda_1$, which is $1.5\%$ at iteration 0, has dropped to $0.2\%$ at iteration 2000. The interpretation is that the tangent kernel *stretches* in the directions of the signal during training.

---

[1]See Appendix A for the extension to vector-valued functions, along with further mathematical details.

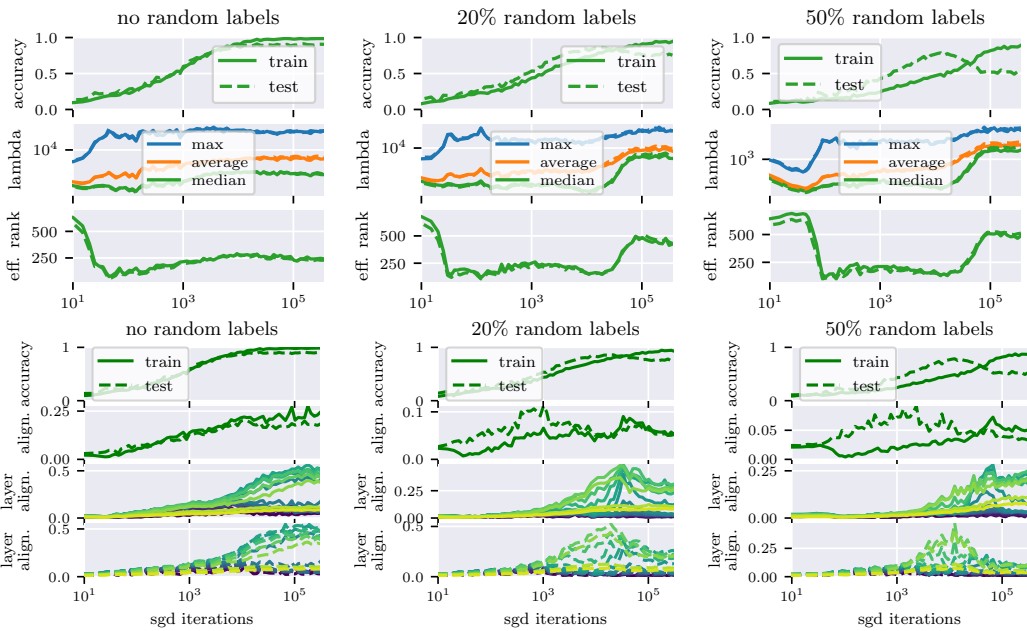

Figure 1: Evolution the *spectrum* and *effective rank* of the tangent kernel (**1st row**) and CKA and layer-wise CKA (**2nd row**) of a VGG19 on CIFAR10 with various ratios of random labels. For layer-wise alignment we map layers to colors sequentially from input layer (**-**), through intermediate layers (**-**), to output layer (**-**).

## 1 Neural Feature Alignment

We run experiments on MNIST [35] and CIFAR10 [33] with standard MLPs, VGG [55] and Resnet [28] architectures, using PyTorch [47] and NNGeometry [3] for efficient evaluation of tangent kernels. In multiclass settings, tangent kernels on $n$ samples carry additional class indices $y \in \{1 \cdots c\}$ and are treated as $nc \times nc$ matrices. We evaluate them on mini-batches (train or test) of size $n = 100$.

**Spectrum Evolution.** We report results (Fig. 1, 1st row) for tangent kernels evaluated on training examples (solid line) and test examples (dashed line). The main take away is an anisotropic increase of the kernel/metric spectrum during training. We quantify spectrum anisotropy through the various **trace ratios** $T_k = \sum_{j<k} \lambda_j / \sum_j \lambda_j$ as measures of the relative importance of the top $k$ eigenvalues ; and using a notion of **effective rank** based on spectral entropy [50] (Longer Version, Appendix D).

We note an important decrease of the effective rank early in training, reaching a phase where only a few top eigenvalues account for most of the trace. This can be observed directly (Fig. 15) from the highlighted (in red) ratios $T_{40}$, $T_{80}$ and $T_{160}$ (Fig. 15), e.g. $T_{80}$ accounting for 50% of the total trace (over 1000 eigenvalues). Remarkably, in the presence of high label noise, the effective rank of the tangent kernel evaluated on *training* examples (anti)-correlates nicely with the *test* accuracy, decreasing or remaining low during the learning phase and rising when overfitting starts. This suggests that the effective rank already provides a good proxy for the network's effective capacity.

**Alignment to class labels.** We investigate the similarity of the tangent features with $Y \in \mathbb{R}^{nc}$ (concatenated one-hot vectors) through the **centered kernel alignment** (CKA) [19, 18] (Appendix D) $\mathrm{CKA}(K_\mathbf{w}, K_Y)$ with the rank-one kernel $K_Y := YY^\top$. Intuitively, it is high when $K_\mathbf{w}$ has low (effective) rank, and is such that the angle between $Y$ and its top eigenspaces is small.[2] Maximizing such an index has been used as a criterion for kernel selection in the literature on learning kernels [18].

We observe (Fig. 1, 2nd row) an increasingly high CKA as training progresses. The trend is similar for other architectures and datasets (Fig. 13 in Appendix E). Interestingly, in the presence of high level noise and during the learning phase, the CKA reaches a much higher value for *test* than for *train* kernels/labels (note that test labels are not randomized). Together with equation 11, this sheds lights on empirical observations that, in the presence of noise, deep networks 'learn patterns first' [5]

---

[2]In the limiting case $\mathrm{CKA}(K, K_Y) = 1$, the features are all aligned with each other and parallel to $Y$.

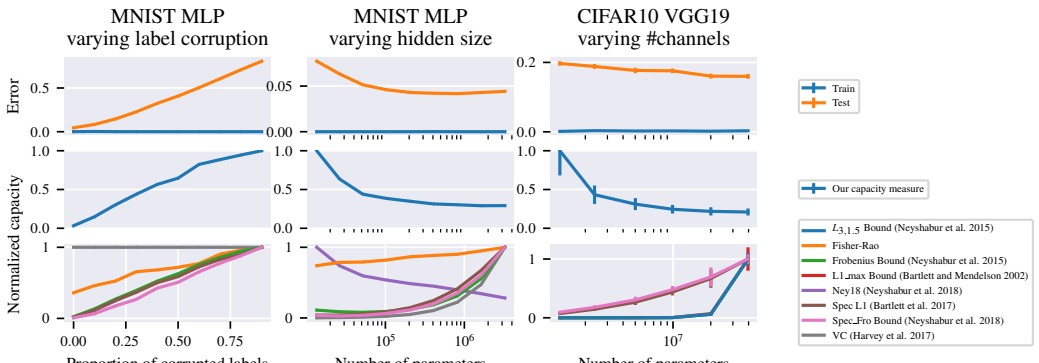

Figure 2: Normalized complexity measures on MNIST with a one hidden layer MLP **(left)** as we increase the hidden layer size, **(center)** for a fixed hidden layer of 256 units as we increase label corruption and **(right)** for a VGG19 on CIFAR10 as we vary the number of channels.

**Hierarchical Alignment.** A key aspect of the generalization question concerns the articulation between learning and memorization, in the presence of noise [61] or difficult examples [51]. In our next experiment, our setup is to augment 10.000 MNIST training examples with 1000 difficult examples of 2 types: (*i*) examples with random labels and (*ii*) examples from the dataset KMNIST [17]. Fig. 6 (Longer Version) shows that the (partial) CKA on the easy examples increases faster (and to a higher value) than that of the difficult examples. This suggests a hierarchy in the adaptation of the kernel, measured by the ratio between both alignments. This aspect of the non-linear dynamics favors a sequentialization of the learning ('easy patterns first') (see [52, 34, 25] for deep linear networks.) Fig 16 (Appendix E) shows that this effect is magnified as depth increases.

## 2 Measuring Complexity

### 2.1 Insights from Linear Models

**Setup.** We restrict here to functions $f_{\mathbf{w}}(\mathbf{x}) = \langle \mathbf{w}, \Phi(\mathbf{x}) \rangle$ linearly parametrized by $\mathbf{w} \in \mathbb{R}^P$. In this setting, (tangent) kernel and geometry are constant. Given $n$ input samples, the $n$ features $\Phi(\mathbf{x}_i) \in \mathbb{R}^P$ yield a $n \times P$ feature matrix $\mathbf{\Phi}$. Our discussion is based on the **Rademacher complexity** showing up in generalization bounds [6]. It depends on the size (or **capacity**) of the function class.

A standard approach for controlling capacity is in terms of the *norm* of the weight vector – usually the $\ell_2$-norm. In general, given any invertible matrix $A \in \mathbb{R}^{P \times P}$, we may consider the norm $\|\mathbf{w}\|_A := \sqrt{\mathbf{w}^\top g_A \mathbf{w}}$ induced by the metric $g_A = AA^\top$. For $M_A > 0$, let $\mathcal{F}_{M_A}^A$ be the subclass of functions $f_{\mathbf{w}}$ such that $\|\mathbf{w}\|_A \leq M_A$. The Rademacher complexity can be bounded as,

$$\widehat{\mathcal{R}}(\mathcal{F}_{M_A}^A) \leq (M_A/n)\|A^{-1}\mathbf{\Phi}^\top\|_{\mathrm{F}} \tag{3}$$

in terms of the Froebenius norm of the *rescaled* feature matrix. This raises the question of which of the norms $\|\cdot\|_A$ provide meaningful capacity measures. Recent works [10, 40] pointed out that the $\ell_2$ norm is not coherently linked with generalization in practice. We discuss this issue in Appendix C.5, illustrating how meaningful norms critically depend on the geometry defined by the features.

**Feature Alignment as Implicit Regularization.** The goal here is to illustrate in a simple setting how an *adaptive* geometry can act as implicit regularizer. In such setting, the idea is to *learn* a rescaling metric at each iteration of our algorithm, using a local version of the bounds (71). We consider functions $f_{\mathbf{w}} = \sum_t \delta f_{\mathbf{w}_t}$ written in terms of a sequence of updates[3] $\delta f_{\mathbf{w}_t}(\mathbf{x}) = \langle \delta \mathbf{w}_t, \Phi(\mathbf{x}) \rangle$ (we set $f_0$ to keep the notation simple), with *local* constraints on the parameter updates:

$$\mathcal{F}_{\boldsymbol{m}}^{\boldsymbol{A}} = \{f_{\mathbf{w}} \colon \mathbf{x} \mapsto \sum_t \langle \delta \mathbf{w}_t, \Phi(\mathbf{x}) \rangle \mid \|\delta \mathbf{w}_t\|_{A_t} \leq m_t\} \tag{4}$$

---

[3]In order to not assume a specific upper bound on the number of iterations, we can think of the updates from an iterative algorithm as an infinite sequence $\{\delta \mathbf{w}_0, \cdots \delta \mathbf{w}_t, \cdots\}$ such that for some $T$, $\delta \mathbf{w}_t = 0$ for all $t > T$.

**Theorem 1** (Complexity of Learning Flows). *Given any sequences $\boldsymbol{A}$ and $\boldsymbol{m}$ of invertible matrices $A_t \in \mathbb{R}^{P \times P}$ and positive numbers $m_t > 0$, we have the bound*

$$\widehat{\mathcal{R}}(\mathcal{F}_{\boldsymbol{m}}^{\boldsymbol{A}}) \leq \sum_t (m_t/n) \|A_t^{-1} \boldsymbol{\Phi}^\top\|_{\mathrm{F}} \tag{5}$$

The same result can be formulated in terms of the sequence of feature maps $\Phi_t = A_t^{-1}\Phi$. By reparametrization invariance, the function class (16) can equivalently be written as $\mathcal{F}_{\boldsymbol{m}}^{\boldsymbol{A}} = \mathcal{F}_{\boldsymbol{m}}^{\boldsymbol{\Phi}}$ where $\boldsymbol{\Phi} = \{\Phi_t\}_t$ and the norm constraints are $\|\tilde{\delta}\mathbf{w}_t\|_2 \leq m_t\}$; then (17) reads

$$\widehat{\mathcal{R}}(\mathcal{F}_{\boldsymbol{m}}^{\boldsymbol{\Phi}}) \leq \sum_t (m_t/n) \|\boldsymbol{\Phi}_t\|_{\mathrm{F}} \tag{6}$$

Thm. 3 suggests to include, at each iteration $t$, a reparametrization step with a suitable matrix $\tilde{A}_t$ giving a low contribution to the bound (17). Applied to gradient descent, this leads to the new update rule below, where the optimization in Step 2 is over a given class of matrices.

SuperNat update ($\tilde{A}_0 = \boldsymbol{I}$, $\Phi_0 = \Phi$, $\boldsymbol{K}_0 = \boldsymbol{K}$):

1. Perform gradient step $\widetilde{\mathbf{w}}_{t+1} \leftarrow \mathbf{w}_t + \delta\mathbf{w}_{\mathrm{GD}}$

2. Find minimizer $\tilde{A}_{t+1}$ of $\|\delta\mathbf{w}_{\mathrm{GD}}\|_{\tilde{A}} \|\tilde{A}^{-1}\boldsymbol{\Phi}_t^\top\|_{\mathrm{F}}$

3. Reparametrize:

$$\mathbf{w}_{t+1} \leftarrow \tilde{A}_{t+1}^\top \widetilde{\mathbf{w}}_{t+1}, \Phi_{t+1} \leftarrow \tilde{A}_{t+1}^{-1}\Phi_t$$

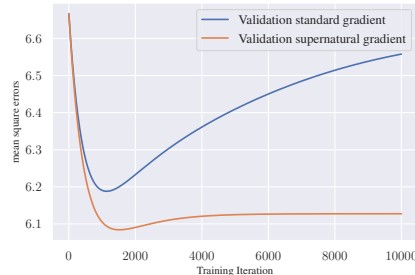

The successive reparametrizations yield a varying feature map $\Phi_t = A_t^{-1}\Phi$ where $A_t = \tilde{A}_0 \cdots \tilde{A}_t$. In the original feature representation $\Phi$, SuperNat amounts to performing natural gradient updates with respect to the local metric $g_{A_t}$; and by construction, we also have $\delta f_{\mathbf{w}_t}(\mathbf{x}) = \langle \delta\mathbf{w}_{\mathrm{GD}}, \Phi_t(\mathbf{x})\rangle$ where $\delta\mathbf{w}_{\mathrm{GD}}$ are standard gradient descent updates in the linear model with feature map $\Phi_t$.

As an example, consider matrices $\tilde{A}_\nu$ acting diagonally in the right singular basis of the feature matrix, i.e by rescaling the singular vectors $\lambda_j \to \lambda_j/\nu_j$. Step 2 can be computed analytically (Longer Version, Prop. 4): up to isotropic rescaling, this yields the update rule $\lambda_{j(t+1)} = |\boldsymbol{u}_j^\top \nabla_{\mathbf{f}_w} L| \lambda_{jt}$ for the singular values of $\boldsymbol{\Phi}_t$. This stretches (resp. contracts) the geometry in directions of large (resp. small) residual $\nabla_{\mathbf{f}_w} L$, thereby increases the alignment of the learned features to the signal. The working hypothesis in this paper, supported by the observations of Section 1, is that in the case of neural networks, such alignment of the features is dynamically induced as an effect of non-linearity.[4]

The plot shows the training curves for a simple model with Gaussian features $\Phi = [\varphi, \varphi_{\mathrm{noise}}] \in \mathbb{R}^{d+1}$ trained to regress $\boldsymbol{y} = \boldsymbol{\varphi} + P_{\mathrm{noise}}(\boldsymbol{\epsilon})$, with Gaussian noise is added in the direction of the noise features. SuperNat identifies dominant features (here $\varphi$) and stretches the metric along them, thereby slowing down and eventually freezing the dynamics in the orthogonal (noise) directions.

## 2.2 A New Complexity Measure for Neural Networks

Equ. (19) provides a bound of the Rademacher complexity for the function classes (16) specified by a fixed sequence of adaptive kernels (see Appendix C.4 for a generalization to the multiclass setting). By extrapolation to the case of non-deterministic sequences of kernels, we propose using

$$\mathcal{C}(f_{\mathbf{w}}) = \sum_t \|\delta\mathbf{w}_t\|_2 \|\boldsymbol{\Phi}_t\|_{\mathrm{F}} \tag{7}$$

where $\boldsymbol{\Phi}_t$ is the tangent feature matrix[5] at training iteration $t$, as a heuristic measure of complexity for neural networks. Following a standard protocol for studying complexity measures, [e.g., 43], Fig. 8 shows its behaviour for MLP on MNIST and VGG19 on CIFAR10 trained with cross entropy loss, with **(left)** fixed architecture and varying level of corruption in the labels and **(right)** varying hidden layer size/number of channels up to 4 millions parameters, against other capacity measures proposed in the recent literature. We observe that it correctly reflects the shape of the generalization gap.

---

[4]For a non-linear model, the updates of the tangent feature take the same form $\Phi_t = \tilde{A}_t^{-1}\Phi_{t-1}$ as above, the difference being that $\tilde{A}_t$ is no longer a matrix but a differential operator, e.g. at first order $A_t = \mathrm{Id} - \delta\mathbf{w}_t^\top \frac{\partial}{\partial\mathbf{w}_t}$.

[5]In terms of tangent kernels, $\|\boldsymbol{\Phi}_t\|_{\mathrm{F}} = \sqrt{\mathrm{Tr}\boldsymbol{K}_t}$ where $\boldsymbol{K}_t$ is the tangent kernel matrix.

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

# Implicit Regularization via Feature Alignment
## (Longer Version)

We approach the problem of implicit regularization in deep learning from a geometrical viewpoint. We highlight a regularization effect induced by a dynamical alignment of the neural tangent features introduced by Jacot et al. [30], along a small number of task-relevant directions. This can be interpreted as a combined feature selection and compression mechanism. By extrapolating a new analysis of Rademacher complexity bounds for linear models, we propose and study a new heuristic measure of complexity which captures this phenomenon, in terms of sequences of tangent kernel classes along the learning trajectories.

## 1 Introduction

One important property of deep neural networks is their ability to generalize well on real data. Surprisingly, this is even true with very high-capacity networks *without explicit regularization* [42, 61, 29]. This seems at odds with the usual understanding of the bias-variance trade-off [24, 41, 11]: highly complex models are expected to overfit the training data and perform poorly on test data [27]. Solving this apparent paradox requires understanding the various learning biases induced by the training procedure, which can act as implicit regularizers [42, 44].

In this paper, we help clarify one such implicit regularization mechanism, by examining the evolution of the *neural tangent features* [30] learned by the network along the optimization paths. Our results can be understood from two complementary perspectives: a *geometric* perspective – the (uncentered) covariance of the tangent features defines a metric on the model function class, akin to the Fisher information metric [e.g., 2]; and a *functional* perspective – through the tangent kernel and its RKHS.

Our main observation, in standard supervised classification settings, is a dynamical alignment of the tangent features along a small number of task-relevant directions during training. We interpret this phenomenon as combining a *feature selection* and a *compression* mechanisms. The intuition motivating this work is that such mechanisms are what allows the model to adapt its capacity to the task, which in turn underpins the generalization abilities of heavily overparametrized models.

Specifically, our main contributions are as follows:

1. Through experiments with various architectures on MNIST and CIFAR10, we give empirical insights on how the tangent features and their kernel adapt to the task during training (Section 3). We observe in particular an increasing similarity with the class labels, e.g. as measured by *centered kernel alignment* (CKA) [19, 18].

2. Drawing upon intuitions from linear models (Section 4.1), we argue that such a dynamical alignment acts as *implicit regularizer*. We motivate a new heuristic complexity measure which captures this phenomenon, and empirically show better correlation with generalization compared to various measures proposed in the recent literature (Section 4).

## 2 Preliminaries

Let $\mathcal{F}$ be a class of functions (e.g a neural network) parametrized by $\mathbf{w} \in \mathbb{R}^P$. We restrict here to *scalar* functions $f_{\mathbf{w}} \colon \mathcal{X} \to \mathbb{R}$ to keep notation light.[6]

**Tangent Features.** We define the **tangent features** as the function gradients w.r.t the parameters,

$$\Phi_{\mathbf{w}}(\mathbf{x}) := \nabla_{\mathbf{w}} f_{\mathbf{w}}(\mathbf{x}) \tag{8}$$

The corresponding kernel $k_{\mathbf{w}}(\mathbf{x}, \tilde{\mathbf{x}}) = \langle \Phi_{\mathbf{w}}(\mathbf{x}), \Phi_{\mathbf{w}}(\tilde{\mathbf{x}}) \rangle$ is the **tangent kernel** [30]. Intuitively, the tangent features govern how small changes in parameter affect the function's outputs,

$$\delta f_{\mathbf{w}}(\mathbf{x}) = \langle \delta \mathbf{w}, \Phi_{\mathbf{w}}(\mathbf{x}) \rangle + O(\|\delta \mathbf{w}\|^2) \tag{9}$$

---

[6]The extension to vector-valued functions, relevant for the multiclass classification setting, is presented in Appendix A, along with more mathematical details.

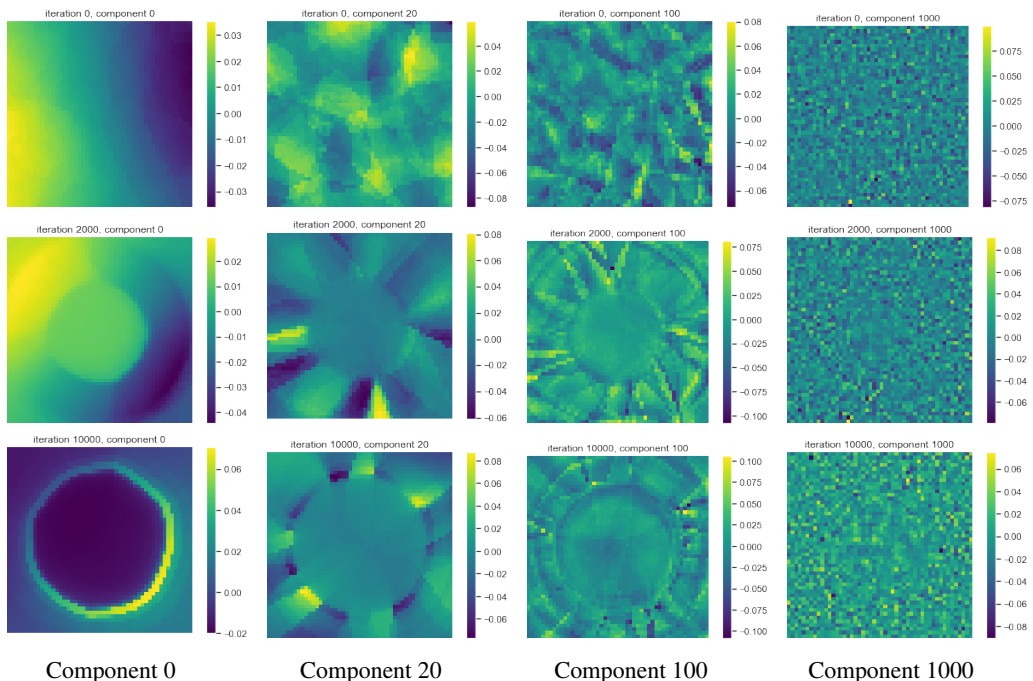

| Component 0 | Component 20 | Component 100 | Component 1000 |

Figure 3: Evolution of eigenfunctions of the tangent kernel, ranked in nonincreasing order of the eigenvalues (**in columns**), at various iterations during training (**in rows**), for the $2d$ Disk dataset. After a number of iterations, we observe modes corresponding to the class structure (e.g. boundary circle) in the top eigenfunctions. Combined with an increasing anistropy of the spectrum (e.g $\lambda_{20}/\lambda_1 = 1.5\%$ at iteration 0, $0.2\%$ at iteration 2000), this illustrates a stretch of the tangent kernel in the directions of the signal.

More formally, the (uncentered) covariance matrix $g_{\mathbf{w}} = \mathbb{E}_{\mathbf{x} \sim \rho}\left[\Phi_{\mathbf{w}}(\mathbf{x})\Phi_{\mathbf{w}}(\mathbf{x})^{\top}\right]$ w.r.t the input distribution $\rho$ acts as a **metric tensor** on $\mathcal{F}$: assuming $\mathcal{F} \subset L^2(\rho)$, this is the metric induced on $\mathcal{F}$ by pullback of the $L^2$ scalar product (see Appendix A). It characterizes the geometry of the function class $\mathcal{F}$.

**Spectral Bias.** The structure of the tangent features impacts the evolution of the function during training. To formalize this, we introduce the covariance eigenvalue decomposition $g_{\mathbf{w}} = \sum_{j=1}^{P} \lambda_{\mathbf{w}j} \boldsymbol{v}_{\mathbf{w}j} \boldsymbol{v}_{\mathbf{w}j}^{\top}$, which summarizes the predominant directions in parameter space. Given $n$ input samples $(\mathbf{x}_i)$ and $\mathbf{f}_{\mathbf{w}} \in \mathbb{R}^n$ the vector of outputs $f_{\mathbf{w}}(\mathbf{x}_i)$, consider gradient descent updates $\delta\mathbf{w}_{\mathrm{GD}} = -\eta\nabla_{\mathbf{w}}L$ for some cost function $L := L(\mathbf{f}_{\mathbf{w}})$. The following elementary result (see Appendix B) shows how the corresponding function updates in the linear approximation (9), $\delta f_{\mathrm{GD}}(\mathbf{x}) := \langle\delta\mathbf{w}_{\mathrm{GD}}, \Phi_{\mathbf{w}}(\mathbf{x})\rangle$, decompose in the **eigenbasis**[7] of the tangent kernel:

$$u_{\mathbf{w}j}(\mathbf{x}) = \frac{1}{\sqrt{\lambda_{\mathbf{w}j}}}\langle\boldsymbol{v}_{\mathbf{w}j}, \Phi_{\mathbf{w}}(\mathbf{x})\rangle \tag{10}$$

**Lemma 2** (Local Spectral Bias). *The function updates decompose as* $\delta f_{GD}(\mathbf{x}) = \sum_{j=1}^{P} \delta f_j u_{\mathbf{w}j}(\mathbf{x})$ *with*

$$\delta f_j = -\eta\lambda_{\mathbf{w}j}(\boldsymbol{u}_{\mathbf{w}j}^{\top}\nabla_{\mathbf{f}_{\mathbf{w}}}L), \tag{11}$$

*where* $\boldsymbol{u}_{\mathbf{w}j} = [u_{\mathbf{w}j}(\mathbf{x}_1), \cdots u_{\mathbf{w}j}(\mathbf{x}_n)]^{\top} \in \mathbb{R}^n$ *and* $\nabla_{\mathbf{f}_{\mathbf{w}}}$ *is the gradient w.r.t the sample outputs.*

This illustrates how, from the point of view of function space, the eigenvalues act as a mode-specific rescaling $\eta\lambda_{\mathbf{w}j}$ of the learning rate. This is a local version of a well-known bias for linear models trained by gradient descent (e.g in linear regression, see Appendix B.2), which prioritizes learning functions within the top eigenspaces of the kernel. Several recent works [12, 9, 60] investigated such

---

[7]The functions $u_{\mathbf{w}j}$, $j \in \{1 \cdots P\}$ form an orthonormal family in $L^2(\rho)$, i.e. $\mathbb{E}_{\mathbf{x} \sim \rho}[u_{\mathbf{w}j}u_{\mathbf{w}j'}] = \delta_{jj'}$, yielding the spectral decomposition $k_{\mathbf{w}}(\mathbf{x}, \tilde{\mathbf{x}}) = \sum_{j=1}^{P} \lambda_{\mathbf{w}j}u_{\mathbf{w}j}(\mathbf{x})u_{\mathbf{w}j}(\tilde{\mathbf{x}})$ of the tangent kernel as an integral operator. Note that kernel and covariance share the same spectrum.

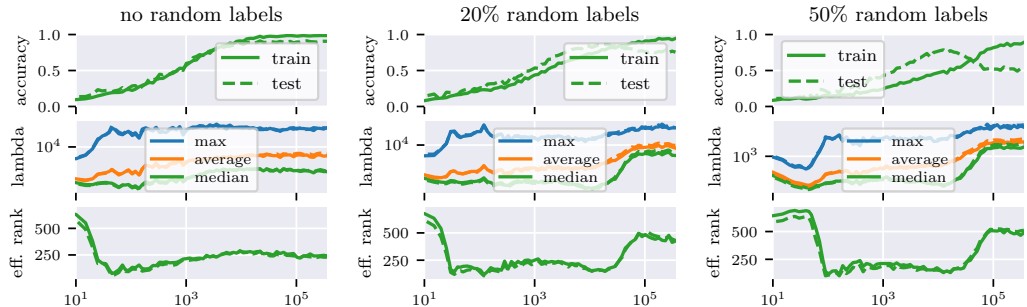

Figure 4: Evolution of tangent kernel spectrum and effective rank of a VGG19 trained by SGD with batch size 100, learning rate 0.01 and momentum 0.9 on CIFAR10 with various ratio of random labels. The small effective rank of the kernel biases the training procedure towards a few top eigenvectors.

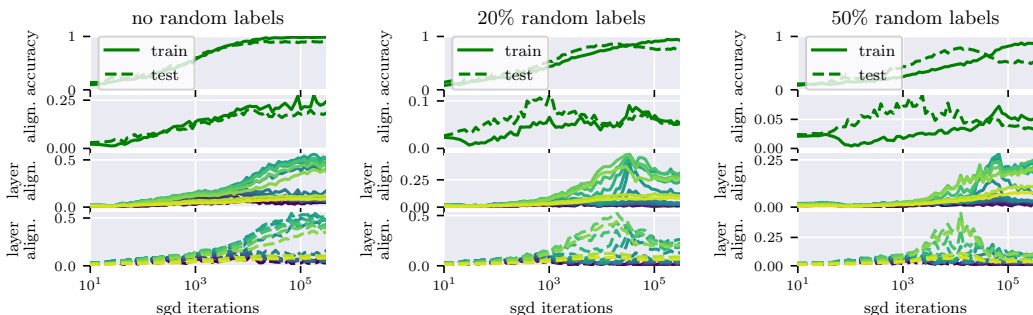

Figure 5: Evolution of kernel alignment and layer-wise kernel alignments of a VGG19 trained by SGD with batch size 100, learning rate 0.01 and momentum 0.9 on CIFAR10 with various ratios of random labels. For layer-wise alignment we map layers to colors sequentially from input layer (-), through intermediate layers (-), to output layer (-). See Figure 13 and 16 in appendix for additional architectures/datasets.

bias for neural networks, in *linearized* regimes where the tangent kernel remains constant during training [30, 20, 1]. As a simple example, for a randomly initialized MLP on 1D uniform data, Fig. 10 (Appendix B) shows an alignment of the tangent kernel eigenfunctions with Fourier modes of increasing frequency, explaining prior empirical observations [48, 59] of a 'spectral bias' towards low-frequency functions.

**Tangent Features Adapt to the Task.** By contrast, our aim in this paper is to highlight and discuss *non-linear* effects, in the (standard) regime where the tangent features and their kernel evolve during training [e.g., 22, 58].

As a first illustration of such effects, Fig. 3 shows visualizations of eigenfunctions of the tangent kernel (ranked in nonincreasing order of the eigenvalues), during training MLP by gradient descent of the binary cross entropy loss, on a simple classification task: $y(\mathbf{x}) = \pm 1$ depending on whether $\mathbf{x} \sim \text{Unif}[-1, 1]^2$ is in the centered disk of radius $\sqrt{2/\pi}$ (details in Appendix E). After a number of iterations, we observe (rotation invariant) modes corresponding to the class structure (e.g. boundary circle) showing up in the *top* eigenfunctions of the learned kernel. We also note an increasing spectrum anisotropy – for example, the ratio $\lambda_{20}/\lambda_1$, which is $1.5\%$ at iteration 0, has dropped to $0.2\%$ at iteration 2000. The interpretation is that the tangent kernel *stretches* along a small number of directions that are highly correlated with the signal during training. We quantify and investigate this alignment effect in more detail below.

## 3   Neural Feature Alignment

In this section, we perform experiments showing a dynamical alignment of the tangent features along a small number of task-relevant directions during training. We show in particular that networks learn

tangent features with increasing similarity with the class labels, as measured by **centered kernel alignment** (CKA) [19, 18]. We interpret this phenomenon as combining both a feature selection and a compression mechanism.

### 3.1 Setup

We run experiments on MNIST [35] and CIFAR10 [33] with standard MLPs, VGG [55] and Resnet [28] architectures, trained by stochastic gradient descent (SGD) with momentum, using cross-entropy loss. We use PyTorch [47] and NNGeometry [3] for efficient evaluation of tangent kernels.

In multiclass settings, tangent kernels evaluated on $n$ samples carry additional class indices $y \in \{1 \cdots c\}$ and thus are $nc \times nc$ matrices, $(\boldsymbol{K}_{\mathbf{w}})_{ij}^{yy'} := k_{\mathbf{w}}(\mathbf{x}_i, y; \mathbf{x}_j, y')$. In all our experiments, we evaluate tangent kernels on mini-batches (either from the train or the test set) of size $n = 100$. For $c = 10$ classes, this yields kernel matrices of size $1000 \times 1000$. We report results obtained from *centered* tangent features $\Phi_{\mathbf{w}}(\mathbf{x}) \to \Phi_{\mathbf{w}}(\mathbf{x}) - \mathbb{E}_{\mathbf{x}}\Phi_{\mathbf{w}}(\mathbf{x})$, though we obtain qualitatively similar results for uncentered features (see plots in Appendix E.2).

### 3.2 Spectrum Evolution

We first investigate the evolution of the tangent kernel *spectrum* for a VGG19 on CIFAR 10, trained with and without label noise (Fig. 4). The main take away is an anisotropic increase of the spectrum during training. We report results for kernels evaluated on training examples (solid line) and test examples (dashed line).[8]

The first observation is a significant *increase* of the spectrum, early in training (note the log scale for the number of iterations). By the time the model reaches $100\%$ training accuracy, the maximum and average eigenvalues have gained more than 2 orders of magnitude.

The second observation is that this evolution is highly *anisotropic*. We quantify spectrum anisotropy using a notion of **effective rank** based on spectral entropy [50]. Given a kernel matrix $\boldsymbol{K}$ in $\mathbb{R}^{r \times r}$ with (strictly) positive eigenvalues $\lambda_1, \cdots, \lambda_r$, let $\mu_j = \lambda_j / \sum_{i=1}^{r} \lambda_j$ be the trace-normalized eigenvalues. The effective rank is defined as $\mathrm{erank} = \exp(H(\boldsymbol{\mu}))$ where $H(\boldsymbol{\mu})$ is the Shannon entropy,

$$H(\boldsymbol{\mu}) = -\sum_{j=1}^{r} \mu_j \log(\mu_j) \tag{12}$$

This effective rank is a real number between 1 and $r$, upper bounded by $\mathrm{rank}(\boldsymbol{K})$, which measures the 'uniformity' of the spectrum through the entropy. We also track the various **trace ratios** $T_k = \sum_{j<k} \lambda_j / \sum_j \lambda_j$ as measures of the relative importance of the top $k$ eigenvalues (see Fig. 15 in Appendix E.3).

We note an important decrease of the effective rank early in training (third row in Fig. 4), reaching a phase where only a few top eigenvalues account for most of the trace. This can be observed directly from the highlighted (in red) ratios $T_{40}$, $T_{80}$ and $T_{160}$ (Fig. 15), e.g. $T_{80}$ accounting for $50\%$ of the total trace (over 1000 eigenvalues). Remarkably, in the presence of high label noise, the effective rank of the tangent kernel evaluated on *training* examples (anti)-correlates nicely with the *test* accuracy, decreasing or remaining low during the learning phase (increase of test accuracy) and rising when overfitting starts (decrease of test accuracy). This suggests that the effective rank of the tangent kernel (and hence that of the metric) might already provide a good proxy for a measure of the effective capacity of the network.

### 3.3 Alignment to class labels

We now include the evolution of the eigenvectors in our analysis. We investigate the similarity of the learned tangent features with the class label through a similarity index called centered kernel alignment. Given two kernel matrices $\boldsymbol{K}$ and $\boldsymbol{K}'$ in $\mathbb{R}^{r \times r}$, it is defined as

$$\mathrm{CKA}(\boldsymbol{K}, \boldsymbol{K}') = \frac{\mathrm{Tr}[\boldsymbol{K}_c \boldsymbol{K}'_c]}{\|\boldsymbol{K}_c\|_F \|\boldsymbol{K}'_c\|_F} \in [0, 1] \tag{13}$$

---

[8]The striking similarity of the plots for train and test kernels suggests that the spectrum of empirical tangent kernels is robust to sampling variations in our setting.

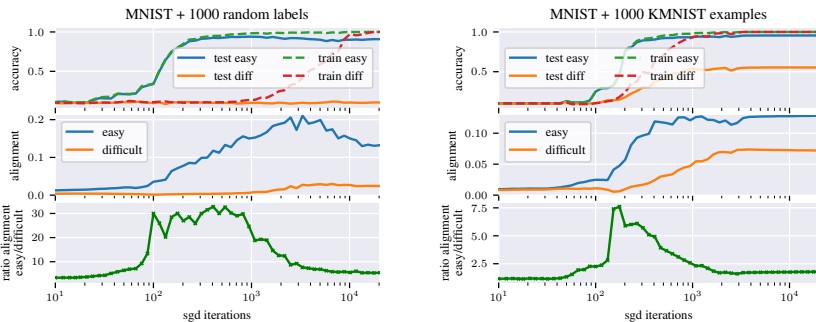

Figure 6: Alignment easy versus difficult: We augment a dataset composed of 10.000 *easy* MNIST examples with 1000 *difficult* examples from 2 different setups: **(left)** 1000 MNIST examples with random label **(right)** 1000 KMNIST examples. We train a MLP with 6 layers of 80 hidden units using SGD with learning rate=0.02, momentum=0.9 and batch size=100. We observe that the NTK aligns faster to the easy examples in the beginning.

where the $c$ subscript denotes the feature centering operation, i.e. $\boldsymbol{K}_c = C\boldsymbol{K}C$ where $C = I_r - \frac{1}{r}\boldsymbol{1}\boldsymbol{1}^T$ is the centering matrix. CKA is a normalized version of the Hilbert-Schmidt Independence Criterion [26] designed as a dependence measure for two sets of features. The normalization by the Froebenius norms makes CKA invariant under isotropic rescaling.

Let $\boldsymbol{Y} \in \mathbb{R}^{nc}$ be the vector resulting from the concatenation of the one-hot label representations $\boldsymbol{Y}_i \in \mathbb{R}^c$ of the $n$ samples. Similarity with the labels is measured through CKA with the rank-one kernel $\boldsymbol{K}_Y := \boldsymbol{Y}\boldsymbol{Y}^\top$. Intuitively, $\mathrm{CKA}(\boldsymbol{K}, \boldsymbol{K}_Y)$ is high when $\boldsymbol{K}$ has low (effective) rank and such that the angle between $\boldsymbol{Y}$ and its top eigenspaces is small.[9] Maximizing such index has been used as a criterion for kernel selection in the literature on learning kernels [18].

In the same setup as in Section 3.2, we observe (Fig. 5 an increasingly high CKA between tangent kernel and the labels as training progresses. The trend is similar for other architectures and datasets (Fig. 13 in Appendix E show CKA plots for MLP on MNIST and Resnets 18 on CIFAR10).

Interestingly, in the presence of high level noise and during the learning phase (increase of test accuracy), the CKA reaches a much higher value for kernels evaluated on test inputs than for kernels evaluated on training inputs (note that test labels are not randomized). Together with equation 11, the alignment of the tangent kernel along clean labels sheds lights on empirical observations that, in the presence of noise, deep networks 'learn patterns first' [5] (see Section 3.4 for additional insights).

We also report the alignments of the *layer-wise* tangent kernels $\boldsymbol{K}_{\mathbf{w}}^\ell$, obtained from the function gradients w.r.t parameters of layer $\ell$. By construction, the tangent kernel is the sum of the layer-wise kernels over all layers of the network, $\boldsymbol{K}_{\mathbf{w}} = \sum_{\ell=1}^{L} \boldsymbol{K}_{\mathbf{w}}^\ell$. We observe a high CKA (reaching more than 0.5), especially for the *intermediate* layers[10], suggesting the key role of depth in the overall alignment of the tangent kernel (see also Section 3.5).

## 3.4 Hierarchical Alignment

A key aspect of the generalization question for deep networks concerns the articulation between learning and memorization, in the presence of noise [61] or difficult examples [51]. Motivated by this, we would like to probe the evolution of the tangent features separately in the directions of both type of examples in such settings. To do so, our strategy is to measure partial CKA on examples from two subsets of the same size in the dataset – one with 'easy' examples, the other with 'difficult' ones. Our setup is to augment 10.000 MNIST training examples with 1000 difficult examples of 2 types: (*i*) examples with random labels and (*ii*) examples from the dataset KMNIST [17]. KMNIST images present similar features than MNIST digits (grayscale handwritten characters) but represent Japanese characters.

---

[9]In the limiting case $\mathrm{CKA}(\boldsymbol{K}, \boldsymbol{K}_Y) = 1$, the features are all aligned with each other and parallel to $Y$.

[10]We were expecting to see a gradually increasing CKA with $\ell$; we do not have any intuitive explanation for the relatively low alignment observed for the very top layers.

The results are shown in Fig. 6. As training progresses, the CKA on the easy examples increases faster (and to a higher value); in the case of the (structured) difficult examples from KMNIST, we observe an increase of the CKA later in training. This demonstrates a hierarchy in the adaptation of the kernel, measured by the ratio between both alignments. From the intuition developed in the paper (see Section 2), this aspect of the non-linear dynamics favors a sequentialization of the learning ('easy patterns first'), a phenomenon analogous to one pointed out in the context of deep linear networks [52, 34, 25].

## 3.5 Ablation

In order to study the influence of depth on alignment and test the robustness to the choice of seeds, we reproduce the experiment of the previous section for MLP with different depths, while varying parameter initialization and minibatch sampling. Our results, shown in Fig 16 (Appendix E), suggest that the alignment effect is magnified as depth increases. We also observe that the ratio of the maximum alignment between easy and difficult examples is increased with depth, but stays high for a smaller number of iterations.

# 4 Measuring Complexity

In this section, drawing upon intuitions from linear models, we illustrate on a simple setting how the alignment effect highlighted in the previous section can act as implicit regularization. We also motivate a new complexity measure for neural networks and compare its correlation to generalization against various measures proposed in the recent literature.

## 4.1 Insights from Linear Models

**Setup.** We restrict here to functions $f_{\mathbf{w}}(\mathbf{x}) = \langle \mathbf{w}, \Phi(\mathbf{x}) \rangle$ linearly parametrized by $\mathbf{w} \in \mathbb{R}^P$. Such function class defines a constant (tangent) kernel and has a constant geometry, as defined in Section 2. Given $n$ input samples, the $n$ features $\Phi(\mathbf{x}_i) \in \mathbb{R}^P$ yield a $n \times P$ feature matrix $\mathbf{\Phi}$.

Our discussion will be based on the **Rademacher complexity**, which shows up in generalization bounds [6]. It measures how well $\mathcal{F}$ correlates with random noise on the sample set $\mathcal{S}$:

$$\widehat{\mathcal{R}}_{\mathcal{S}}(\mathcal{F}) = \mathbb{E}_{\boldsymbol{\sigma} \in \{\pm 1\}^n} \left[ \sup_{f \in \mathcal{F}} \frac{1}{n} \sum_{i=1}^n \sigma_i f(\mathbf{x}_i) \right] \tag{14}$$

The Rademacher complexity depends on the size (or **capacity**) of the class $\mathcal{F}$. Constraints on the capacity, such as those induced by some implicit bias of the training algorithm, can reduce the Rademacher complexity and lead to sharper generalization bounds.

A standard approach for controlling capacity is in terms of the *norm* of the weight vector – usually the $\ell_2$-norm. In general, given any invertible matrix $A \in \mathbb{R}^{P \times P}$, we may consider the norm $\|\mathbf{w}\|_A := \sqrt{\mathbf{w}^\top g_A \mathbf{w}}$ induced by the metric $g_A = AA^\top$. For $M_A > 0$, let $\mathcal{F}_{M_A}^A$ be the subclass of functions $f_{\mathbf{w}}$ such that $\|\mathbf{w}\|_A \leq M_A$. A direct extension of standard bounds for the Rademacher complexity (see Appendix C) yields,

$$\widehat{\mathcal{R}}_{\mathcal{S}}(\mathcal{F}_{M_A}^A) \leq (M_A/n)\|A^{-1}\mathbf{\Phi}^\top\|_{\mathrm{F}} \tag{15}$$

where $\|A^{-1}\mathbf{\Phi}^\top\|_{\mathrm{F}}$ is the Froebenius norm of the *rescaled* feature matrix.

This freedom in the choice of rescaling matrix $A$, due to linear reparametrization invariance, raises the question of which of the norms $\|\cdot\|_A$ provides meaningful measures of the model's capacity. Recent works [10, 40] pointed out that using $\ell_2$ norm is not coherently linked with generalization in practice. We discuss this issue in Appendix C.5, illustrating how meaningful norms critically depend on the geometry defined by the features.[11]

---

[11] Analysis of the relation between capacity and feature geometry can be traced back to early work on kernel methods [53]

SuperNat update ($\tilde{A}_0 = \boldsymbol{I}$, $\Phi_0 = \Phi$, $\boldsymbol{K}_0 = \boldsymbol{K}$):

1. Perform gradient step $\widetilde{\mathbf{w}}_{t+1} \leftarrow \mathbf{w}_t + \delta \mathbf{w}_{\mathrm{GD}}$

2. Find minimizer $\tilde{A}_{t+1}$ of $\|\delta \mathbf{w}_{\mathrm{GD}}\|_{\tilde{A}} \|\tilde{A}^{-1} \boldsymbol{\Phi}_t^\top\|_{\mathrm{F}}$

3. Reparametrize:

$$\mathbf{w}_{t+1} \leftarrow \tilde{A}_{t+1}^\top \widetilde{\mathbf{w}}_{t+1}, \, \Phi_{t+1} \leftarrow \tilde{A}_{t+1}^{-1} \Phi_t$$

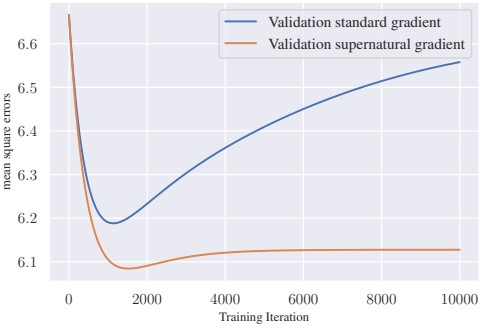

Figure 7: **(left)** SuperNat algorithm and **(right)** validation curves obtained with standard and SuperNat gradient descent, on the noisy linear regression problem. At each iteration, SuperNat identifies dominant features and stretches the kernel along them, thereby slowing down and eventually freezing the learning dynamics in the noise direction. This naturally yields better generalization than standard gradient descent on this problem.

### 4.1.1 Feature Alignment as Implicit Regularization

The goal here is to illustrate in a simple setting how an *adaptive* geometry along optimization trajectories can act as an implicit regularizer. In such setting, the idea is to *learn* a rescaling metric at each iteration of our algorithm, using a local version of the bounds (71).

**Complexity of Learning Flows.** Since we are interested in functions $f_{\mathbf{w}}$ that result from an iterative algorithm, we can assume they are written as $f_{\mathbf{w}} = f_0 + \sum_t \delta f_{\mathbf{w}_t}$ in terms of a sequence of updates $\delta f_{\mathbf{w}_t}(\mathbf{x}) = \langle \delta \mathbf{w}_t, \Phi(\mathbf{x}) \rangle$.[12] We set $f_0 = 0$ to keep the notation simple. Instead of considering classes of functions with direct constraints on the parameter, we consider functions resulting from a learning flow with *local* constraints on the parameter *updates*:

$$\mathcal{F}_{\boldsymbol{m}}^{\boldsymbol{A}} = \{f_{\mathbf{w}} \colon \mathbf{x} \mapsto \sum_t \langle \delta \mathbf{w}_t, \Phi(\mathbf{x}) \rangle \mid \|\delta \mathbf{w}_t\|_{A_t} \leq m_t\} \tag{16}$$

The result (71) extends as follows.

**Theorem 3** (Complexity of Learning Flows). *Given any sequences $\boldsymbol{A}$ and $\boldsymbol{m}$ of invertible matrices $A_t \in \mathbb{R}^{P \times P}$ and positive numbers $m_t > 0$, we have the bound*

$$\widehat{\mathcal{R}}_{\mathcal{S}}(\mathcal{F}_{\boldsymbol{m}}^{\boldsymbol{A}}) \leq \sum_t (m_t/n) \|A_t^{-1} \boldsymbol{\Phi}^\top\|_{\mathrm{F}} \tag{17}$$

Equ. 17 provides us with bounds written in terms of local contributions at each iteration $t$. Note that the same result can be formulated in terms of the sequence of feature maps $\Phi_t = A_t^{-1} \Phi$. By reparametrization invariance, the function class (16) can equivalently be written as $\mathcal{F}_{\boldsymbol{m}}^{\boldsymbol{A}} = \mathcal{F}_{\boldsymbol{m}}^{\boldsymbol{\Phi}}$ where $\boldsymbol{\Phi} = \{\Phi_t\}_t$ and

$$\mathcal{F}_{\boldsymbol{m}}^{\boldsymbol{\Phi}} = \{f_{\mathbf{w}} \colon \mathbf{x} \mapsto \sum_t \langle \tilde{\delta} \mathbf{w}_t, \Phi_t(\mathbf{x}) \rangle \mid \|\tilde{\delta} \mathbf{w}_t\|_2 \leq m_t\} \tag{18}$$

In this formulation, the result (17) reads:

$$\widehat{\mathcal{R}}_{\mathcal{S}}(\mathcal{F}_{\boldsymbol{m}}^{\boldsymbol{\Phi}}) \leq \sum_t (m_t/n) \|\boldsymbol{\Phi}_t\|_{\mathrm{F}} \tag{19}$$

**Optimizing the Feature Scaling.** To obtain learning flows with low complexity, Thm. 3 suggests to include, at each iteration $t$, a reparametrization step with a suitable matrix $\tilde{A}_t$ giving a low contribution to the bound (17). Applied to gradient descent (GD), this leads to a new update rule sketched as in Fig 7 (left), where the optimization in Step 2 is over a given class of reparametrization matrices. As an example, we consider the class of matrices $A_\nu$ acting diagonally in the right singular basis of the feature matrix $\boldsymbol{\Phi} = \sum_{j=1}^n \sqrt{\lambda_j} \boldsymbol{u}_j \boldsymbol{v}_j^\top$; which amounts to rescaling the singular vector $\lambda_j \to \lambda_j/\nu_j$.

---

[12]In order to not assume a specific upper bound on the number of iterations, we can think of the updates from an iterative algorithm as an infinite sequence $\{\delta \mathbf{w}_0, \cdots \delta \mathbf{w}_t, \cdots\}$ such that for some $T$, $\delta \mathbf{w}_t = 0$ for all $t > T$.

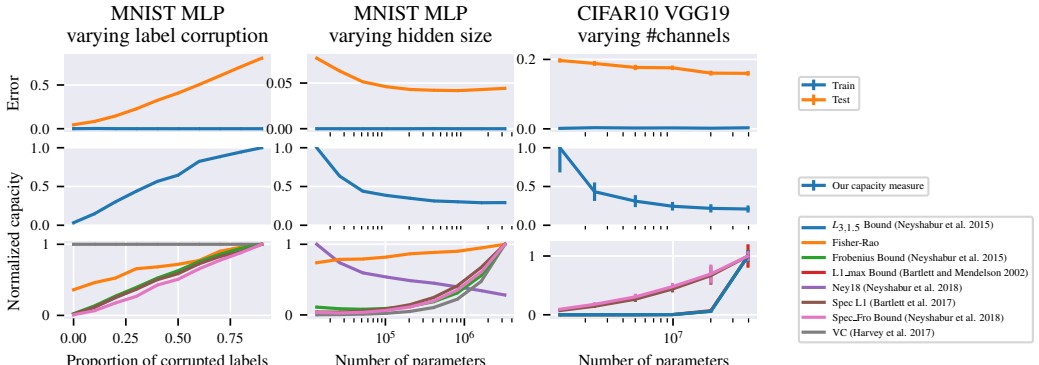

Figure 8: Complexity measures on MNIST with a one hidden layer MLP **(left)** as we increase the hidden layer size, **(center)** for a fixed hidden layer of 256 units as we increase label corruption and **(right)** for a VGG19 on CIFAR10 as we vary the number of channels. All networks are trained until cross-entropy loss reaches 0.01. Our proposed complexity measure and the one proposed by Neyshabur et al. 2018 are the only ones to correctly reflect the shape of the generalization gap.

**Proposition 4.** *For the class of rescaling matrices $A_\nu$ defined above, any minimizer in Step 2 in Fig 7, where $\delta \mathbf{w}_{GD} = -\eta \nabla_\mathbf{w} L$ is a GD updates w.r.t a loss $L$, takes the form*

$$\nu_{jt}^* = \kappa \frac{1}{|\boldsymbol{u}_j^\top \nabla_{\mathbf{f}_w} L|} \tag{20}$$

*where $\nabla_{\mathbf{f}_w}$ denotes the gradient w.r.t $f_\mathbf{w} := [f_\mathbf{w}(\mathbf{x}_1), \cdots f_\mathbf{w}(\mathbf{x}_n)]^\top$, for some constant $\kappa > 0$.*

The successive reparametrizations yield a varying feature map $\Phi_t = A_t^{-1} \Phi$ where $A_t = \tilde{A}_0 \cdots \tilde{A}_t$. In the original representation $\Phi$, SuperNat amounts to natural gradient descent with respect to the local metric $g_{A_t} = A_t A_t^\top$. In the context of Proposition 4, this yields the following update rule, up to isotropic rescaling, for the singular values of $\mathbf{\Phi}_t$:

$$\lambda_{j(t+1)} = |\boldsymbol{u}_j^\top \nabla_{\mathbf{f}_w} L| \lambda_{jt} \tag{21}$$

In this illustrative setting, we see how the feature map (or kernel) adapts to the task, by stretching (resp. contracting) its geometry in directions $\boldsymbol{u}_j$ along which the residual $\nabla_{\mathbf{f}_w} L$ has large (resp. small) components. Intuitively, if a large component $|\boldsymbol{u}_j^\top \nabla_{\mathbf{f}_w} L|$ corresponds to signal and a small one $|\boldsymbol{u}_k^\top \nabla_{\mathbf{f}_w} L|$ corresponds to noise, then the ratio $\lambda_{jt}/\lambda_{kt}$ of singular values gets rescaled by the signal-to-noise ratio, thereby increasing the alignment of the learned feature matrix to the signal.

Fig 7 in Appendix **??** (right) shows results for the following regression setup. We consider Gaussian features $\Phi = [\varphi, \varphi_{\text{noise}}] \in \mathbb{R}^{d+1}$ where $\varphi \sim \mathcal{N}(0, 1)$ and $\varphi_{\text{noise}} \sim \mathcal{N}(0, \frac{1}{d} I_d)$. Given $n$ training features, we assume the label vector takes the form $\boldsymbol{y} = \boldsymbol{\varphi} + P_{\text{noise}}(\boldsymbol{\epsilon})$, where Gaussian noise $\boldsymbol{\epsilon} \sim \mathcal{N}(0, \sigma^2 I_n)$ is projected onto the noise features through $P_{\text{noise}} = \boldsymbol{\varphi}_{\text{noise}} \boldsymbol{\varphi}_{\text{noise}}^\top$. The model is trained by gradient descent of the mean square loss and its SuperNat variant, where Step 2 uses the analytical solution of Proposition 4. We set $d = 10, \sigma^2 = 0.1$ and use $n = 50$ training points. At each iteration, SuperNat identifies dominant features (feature selection, here $\varphi$) and stretches the metric along them, thereby slowing down and eventually freezing the dynamics in the orthogonal (noise) directions (compression).

## 4.2 A New Complexity Measure for Neural Networks

Equ. (19) provides a bound of the Rademacher complexity for the function classes (16) specified by a fixed sequence of adaptive kernels (see Appendix C.4 for a generalization to the multiclass setting). By extrapolation to the case of non-deterministic sequences of kernels, we propose using

$$\mathcal{C}(f_\mathbf{w}) = \sum_t \|\delta \mathbf{w}_t\|_2 \|\mathbf{\Phi}_t\|_\text{F} \tag{22}$$

where $\mathbf{\Phi}_t$ is the tangent feature matrix[13] at training iteration $t$, as a heuristic measure of complexity for neural networks. Following a standard protocol for studying complexity measures, [e.g., 43], Fig. 8 shows its behaviour for MLP on MNIST and VGG19 on CIFAR10 trained with cross entropy loss, with **(left)** fixed architecture and varying level of corruption in the labels and **(right)** varying hidden layer size/number of channels up to 4 millions parameters, against other capacity measures proposed in the recent literature. We observe that it correctly reflects the shape of the generalization gap.

# 5   Related Work

**Capacity and Geometry.** In the context of linear models, analysis of the relation between capacity and feature geometry can be traced back to early work on kernel methods (Schölkopf et al. [53]), leading to data-dependent error bounds in terms of the eigenvalues of the kernel Gram matrix (Schölkopf et al. [54]). Recent analysis of the minimum norm interpolators in overparametrized linear regression emphasized the impact of feature geometry – through the spectrum of the data covariance – on generalization performance [8, 39].

Specifically, these works illustrate the key role of *feature anisotropy*, combined to a high correlation of the few dominant features with the signal [see also e.g. 13], in the generalization performance [39]. Both feature anisotropy and high correlation of the dominant features are conditions for a high alignment between kernel and labels. Our results in this paper emphasizes the key role of the training dynamics in favouring such conditions.

**Generalization Measures.** There has been a large body of work on generalization measures for neural networks (see Jiang et al. [31] and references therein), some of which theoretically motivated by norm or margin based bounds (e.g Neyshabur et al. [45], Bartlett et al. [7]). Liang et al. [37] proposed using the Fisher-Rao norm to measure capacity in a geometrical invariant manner. Our approach aims at taking into account the geometry along the whole optimization trajectories. Closely related perspectives in the recent literature are the notion of stiffness [21] and coherent gradients [15], tied to the structure of tangent kernels for the loss class.

**Spectral Bias and Tangent Kernels**. A recent line of work on the so-called *spectral bias* [48, 59], relying on Fourier analysis, suggested that neural networks prioritize learning the lowest complexity components of the data during training. In *linearized* regimes where the training dynamics can be described by a fixed kernel [30, 20, 16], this can be understood in terms of the standard learning bias along the kernel principal components in linear regression [4, 9, 14]. Several other works [12, 9, 60] investigated implicit bias of neural networks through a spectral analysis in such regimes. In this paper, we highlight and discuss *non-linear* effects, in the feature learning regime where the tangent kernel evolves during training [22, 58].

Independent concurrent works highlight alignment phenomena similar to the one we study here [32, 46]. We offer various complementary empirical insights, and frame the alignment mechanism from the point of view of implicit regularization.

# 6   Conclusion

Through experiments on modern architectures, we highlighted an alignment effect of the tangent features and their kernel along a small number of task-dependent directions, quantified by centered kernel alignment. We interpret this phenomenon as combining a *feature selection* mechanism and a *compression* of the model around the dominant features.

We argued that such a dynamical alignment can act as implicit regularization. By extrapolating Rademacher complexity theory from linear models to learning flows, we introduced a new heuristic complexity measure for neural networks, and showed that it correlates with the generalization gap when varying the number of parameters, and when increasing the proportion of corrupted labels.

The results of this paper open several avenues for further investigation. The type of complexity measure we propose suggests a principled way to rescale the geometry in which to perform gradient descent [56, 44]. Whether a procedure such as SuperNat, which optimizes a preconditioning matrix so

---

[13]In terms of tangent kernels, $\|\mathbf{\Phi}_t\|_\mathrm{F} = \sqrt{\mathrm{Tr}\,\boldsymbol{K}_t}$ where $\boldsymbol{K}_t$ is the tangent kernel matrix.

as to minimize a generalization bound[14], can produce meaningful practical results for neural networks, remains to be seen.

One of the consequences one can expect from alignment effect highlighted here is to encourage learning from a small number of highly predictive features. While this feature selection ability might explain in part the performance of neural networks on a range of supervised tasks, it may also might underpin their notorious sensitivity to spurious correlations [51] and weakness to generalize out-of-distribution [23]. Resolving this tension is a fascinating challenge.

## A  Geometry and Tangent Kernels

We describe in more formal detail the notion of geometry we consider in the paper for parametric function classes. Formally, specifying such a geometry relies on a choice a distance measure or metric on the function space, which is then pulled back to parameter space. We will consider general classes of predictors:

$$\mathcal{F} = \{f_{\mathbf{w}} \colon \mathcal{X} \to \mathbb{R}^c \mid \mathbf{w} \in \mathcal{W}\}, \tag{23}$$

where the parameter space $\mathcal{W}$ is a finite dimensional manifold of dimension $P$ (typically $\mathbb{R}^P$). For multiclass classification, $f_{\mathbf{w}}$ outputs a score $f_{\mathbf{w}}(\mathbf{x})[y]$ for each class $y \in \{1 \cdots c\}$. Each function can also be viewed as a scalar function on $\mathcal{X} \times \mathcal{Y}$ where $\mathcal{Y} = \{1 \cdots c\}$ is the set of classes.

We assume that $\mathbf{w} \to f_{\mathbf{w}}$ is a smooth mapping from $\mathcal{W}$ to $L^2(\rho, \mathbb{R}^c)$, where $\rho$ is some input data distribution. The inclusion $\mathcal{F} \subset L^2(\rho, \mathbb{R}^c)$ equips $\mathcal{F}$ with the $L^2$ scalar product and corresponding norm:

$$\langle f, g \rangle_\rho := \mathbb{E}_{\mathbf{x} \sim \rho}[f(\mathbf{x})^\top g(\mathbf{x})], \qquad \|f\|_\rho := \sqrt{\langle f, f \rangle_\rho} \tag{24}$$

The parameter space $\mathcal{W}$ inherits a **metric tensor** $g_{\mathbf{w}}$ by pull-back of the scalar product $\langle f, g \rangle_\rho$ on $\mathcal{F}$. That is, given $\boldsymbol{\zeta}, \boldsymbol{\xi} \in \mathcal{T}_{\mathbf{w}}\mathcal{W} \cong \mathbb{R}^P$ on the tangent space at $\mathbf{w}$,

$$g_{\mathbf{w}}(\boldsymbol{\zeta}, \boldsymbol{\xi}) = \langle \partial_{\boldsymbol{\zeta}} f_{\mathbf{w}}, \partial_{\boldsymbol{\xi}} f_{\mathbf{w}} \rangle_\rho \tag{25}$$

where $\partial_{\boldsymbol{\zeta}} f_{\mathbf{w}} = \langle df_{\mathbf{w}}, \boldsymbol{\zeta} \rangle$ is the directional derivative in the direction of $\boldsymbol{\zeta}$. Concretely, in a given basis of $\mathbb{R}^P$, the metric is represented by the matrix of gradient second moments:

$$(g_{\mathbf{w}})_{pq} = \mathbb{E}_{\mathbf{x} \sim \rho} \left[ \left( \frac{\partial f_{\mathbf{w}}(\mathbf{x})}{\partial w_p} \right)^\top \frac{\partial f_{\mathbf{w}}(\mathbf{x})}{\partial w_q} \right] \tag{26}$$

where $w_p$, $p = 1, \cdots P$ denote the parameter coordinates. The metric shows up by spelling out the line element $ds^2 := \|df_{\mathbf{w}}\|_\rho^2$, since we have,

$$\|df_{\mathbf{w}}\|_\rho^2 = \sum_{p,q=1}^{P} \langle \frac{\partial f_{\mathbf{w}}}{\partial w_p} dw_p, \frac{\partial f_{\mathbf{w}}}{\partial w_q} dw_q \rangle_\rho = \sum_{p,q=1}^{P} (g_{\mathbf{w}})_{pq}\, dw_p dw_q \tag{27}$$

This geometry has a dual description in function space in terms of *kernels*. The idea is to view the differential at each $\mathbf{w}$ as a map $df_{\mathbf{w}} \colon \mathcal{X} \times \mathcal{Y} \to \mathcal{T}_{\mathbf{w}}^*\mathcal{W} \cong \mathbb{R}^p$ defining (joined) features in the (co)tangent space. Thus, in a given basis, the **tangent features** are given by the function derivatives with respect to the parameters

$$\Phi_{w_p}(\mathbf{x})[y] := \frac{\partial f_{\mathbf{w}}(\mathbf{x})[y]}{\partial w_p} \tag{28}$$

The tangent feature map $\Phi_{\mathbf{w}}$ can be viewed as a function mapping each pair $(\mathbf{x}, y)$ to a vector in $\mathbb{R}^P$. It defines the so-called **tangent kernel** through the Euclidean dot product in $\mathbb{R}^P$:

$$k_{\mathbf{w}}(\mathbf{x}, y; \tilde{\mathbf{x}}, y') = \sum_{p=1}^{P} \frac{\partial f_{\mathbf{w}}(\mathbf{x})[y]}{\partial w_p} \frac{\partial f_{\mathbf{w}}(\tilde{\mathbf{x}})[y']}{\partial w_p} \tag{29}$$

Given $n$ input samples $\mathbf{x}_1, \cdots \mathbf{x}_n$, we represent the sample output scores $f_{\mathbf{w}}(\mathbf{x}_i)[y]$ as flattened in a single vector $\mathbf{f}_{\mathbf{w}} \in \mathbb{R}^{nc}$ and the tangent features $\Phi_{w_p}(\mathbf{x}_i)[y]$ as a $nc \times P$ matrix $\mathbf{\Phi}_{\mathbf{w}}$. Using this notation, (26) and (29) yield the sample covariance $P \times P$ matrix and kernel (Gram) $nc \times nc$ matrix:

$$\boldsymbol{G}_{\mathbf{w}} = \mathbf{\Phi}_{\mathbf{w}}^\top \mathbf{\Phi}_{\mathbf{w}}, \quad \boldsymbol{K}_{\mathbf{w}} = \mathbf{\Phi}_{\mathbf{w}} \mathbf{\Phi}_{\mathbf{w}}^\top \tag{30}$$

---

[14]See the recent work by [57] for further empirical investigations of this problem in the context of linear models.

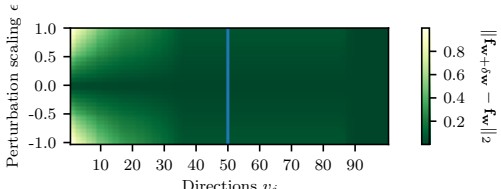

Figure 9: Variations of $\mathbf{f_w}$ (evaluated on a test set) when perturbing the parameters in the directions given by the right singular vectors of the Jacobian (first 50 directions) or in randomly sampled directions (last 50 directions) on a VGG11 network trained for 10 epochs on CIFAR10. We observe that perturbations in most directions have almost no effect, except in those aligned with the top singular vectors.

The eigenvalue decompositions of $G_\mathbf{w}$ and $K_\mathbf{w}$ follow from the (SVD) of $\Phi_\mathbf{w}$. Assuming $P > nc$, we can write this SVD by indexing the singular values by a pair $J = (i, y)$ with $i = 1, \cdots n$ and $y = 1 \cdots c$ as $\Phi_\mathbf{w} = \sum_{J=1}^{nc} \sqrt{\lambda_J} u_J v_J^\top$. Such decompositions summarize the predominant directions both in parameter and feature space, in the neighborhood of $\mathbf{w}$. Indeed, A small variation $\delta\mathbf{w}$ around $\mathbf{w}$ induces the first order variation $\delta\mathbf{f_w}$ of the function given by

$$\delta\mathbf{f_w} := \Phi_\mathbf{w}\delta\mathbf{w} = \sum_{J=1}^{nc} \sqrt{\lambda_J}(v_J^T \delta\mathbf{w})u_J \tag{31}$$

Fig.9 illustrates this 'hierarchy' for a VGG11 network [55] trained for 10 epoches on CIFAR10 [33]. We observe that perturbations in most directions have almost no effect, except in those aligned with the top singular vectors. This is reflected by a strong anisotropy of tangent kernel spectrum.

# B    Spectral Bias

We spell out some more detail for the content of Section 2.

## B.1    Proof of Lemma 2

We consider parameter updates $\delta\mathbf{w}_{\mathrm{GD}} := -\eta\nabla_\mathbf{w}L$ for gradient descent w.r.t the loss $L$. Using the chain rule, we can also write,

$$\delta\mathbf{w}_{\mathrm{GD}} = -\eta\Phi_\mathbf{w}^\top(\nabla_{\mathbf{f_w}}L) \tag{32}$$

**Theorem 5** (Lemma 2 restated)**.** *The gradient descent function updates in first order Taylor approximation, $\delta f_{GD}(\mathbf{x}) := \langle \delta\mathbf{w}_{GD}, \Phi_\mathbf{w}(\mathbf{x})\rangle$, decompose as,*

$$\delta f_{GD}(\mathbf{x}) = \sum_{j=1}^{n} \delta f_j \tilde{u}_{\mathbf{w}j}(\mathbf{x}), \qquad \delta f_j = -\eta\lambda_{\mathbf{w}j}(u_{\mathbf{w}j}^\top \nabla_{\mathbf{f_w}}L) \tag{33}$$

*in terms of the kernel principal components $\tilde{u}_{\mathbf{w}j}$ defined by (10).*

*Proof.* This follows immediately from (32), the SVD of $\Phi_\mathbf{w}$, and the definition (10):

$$\delta f_{\mathrm{GD}}(\mathbf{x}) = -\eta\langle(\nabla_{\mathbf{f_w}}L)^\top \Phi_\mathbf{w}, \Phi_\mathbf{w}(\mathbf{x})\rangle = \sum_{j=1}^{n} \delta f_j \tilde{u}_{\mathbf{w}j}(\mathbf{x}) \tag{34}$$

$\square$

## B.2    The case of linear regression

In this case $L = \frac{1}{2}\|\mathbf{f_w} - y\|^2$ with $f_\mathbf{w} = \langle\mathbf{w}, \Phi(\mathbf{x})\rangle$ (setting of Section 4.1), we can make the 'spectral bias' more explicit. A straightforward consequence of (9) is that the linear system governing the training dynamics in function space decouple in the basis of kernel principal components.

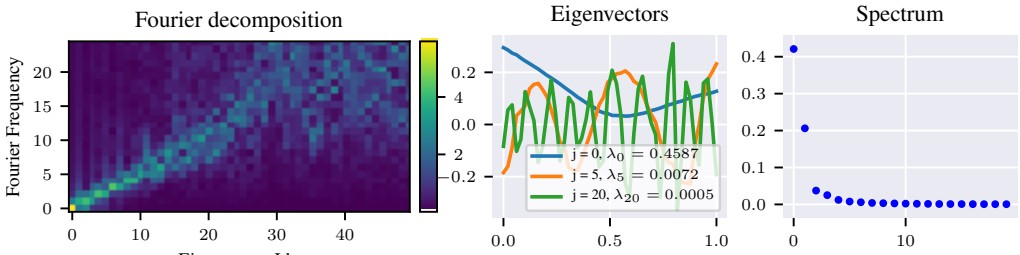

Figure 10: Eigendecomposition of the tangent kernel matrix of a random 6-layer deep 256-unit wide MLP on 1D uniform data (50 equally spaced points in $[0,1]$). **(left)** Fourier decomposition ($y$-axis for frequency, colorbar for magnitude) of each eigenvector ($x$-axis). We observe that eigenvectors with increasing index $j$ correspond to modes with increasing Fourier frequency. **(middle)** Plot of the $j$-th eigenvectors with $j \in \{0, 5, 20\}$ and **(right)** distribution of eigenvalues ranked in nonincreasing order. We note the fast decay (e.g $\lambda_{10}/\lambda_1 \approx 4\%$).

624   Gradient descent yields the function iterates,

$$f_{\mathbf{w}_t} = f_{\mathbf{w}^*} + (\mathrm{id} - \eta K)^t (f_{\mathbf{w}_0} - f_{\mathbf{w}^*}) \tag{35}$$

625   where id is the identity map and $K$ is the operator acting on functions as $(Kf)(\mathbf{x}) =$
626   $\sum_{i=1}^n k(\mathbf{x}, \mathbf{x}_i) f(\mathbf{x}_i)$ in terms of the kernel $k(\mathbf{x}, \tilde{\mathbf{x}}) = \langle \Phi(\mathbf{x}), \Phi(\tilde{\mathbf{x}}) \rangle$.

627   *Proof.* The updates (32) induce the functional updates $\delta f_{\mathrm{GD}} = f_{\mathbf{w}_{t+1}} - f_{\mathbf{w}_t}$ given by

$$\delta f_{\mathrm{GD}}(\mathbf{x}) = -\eta \sum_{i=1}^n k(\mathbf{x}, \mathbf{x}_i)(f_{\mathbf{w}_t}(\mathbf{x}_i) - \boldsymbol{y}_i) \tag{36}$$

628   Substituting $\boldsymbol{y}_i = f_{\mathbf{w}^*}(\mathbf{x}_i)$ gives $f_{\mathbf{w}_{t+1}} - f_{\mathbf{w}^*} = (\mathrm{id} - \eta K)(f_{\mathbf{w}_t} - f_{\mathbf{w}^*})$. Equ. 35 follows by
629   induction. $\qquad\square$

630   The operator $K$ has eigenvalues $\lambda_1, \cdots, \lambda_n$ with eigenfunctions $\tilde{u}_j(\mathbf{x})$ given by (10).

631   *Proof.* We can write $\tilde{u}_j(\mathbf{x}) = \sum_{i=1}^n k(\mathbf{x}, \mathbf{x}_i) u_{ji}$ where $\boldsymbol{u}_j = [u_{j1} \cdots u_{jn}]^\top$ are the eigenvectors of
632   $\boldsymbol{K}$. Observe that $(K\tilde{u}_j)(\mathbf{x}) = \sum_{i=1}^n k(\mathbf{x}, \mathbf{x}_i)(\boldsymbol{K}\boldsymbol{u}_j)_i = \sum_{i=1}^n k(\mathbf{x}, \mathbf{x}_i)(\lambda_j u_{ji}) = \lambda_j \tilde{u}_j$. Conversely,
633   if $\lambda$ is an eigenvalue of $K$ with eigenfunction $\tilde{u}$, consider the vector $\boldsymbol{u} = [\tilde{u}(\mathbf{x}_i) \cdots \tilde{u}(\mathbf{x}_n)]^\top$. Since
634   $\lambda u_i = \tilde{u}(\mathbf{x}_i) = (K\tilde{u})(\mathbf{x}_i) = (\boldsymbol{K}\boldsymbol{u})_i$, $\boldsymbol{u}$ is an eigenvector of $\boldsymbol{K}$ and $\lambda$ is one of the $\lambda_j$. $\qquad\square$

635   [Spectral Bias for Linear Regression] By initializing $\mathbf{w}_0 = \boldsymbol{\Phi}^\top \boldsymbol{\alpha}_0$ in the span of the features, the
636   function iterates in Equ.35 uniquely decompose as $f_{\mathbf{w}_t}(\mathbf{x}) = \sum_{j=1}^n f_{jt} \tilde{u}_j(\mathbf{x})$ with

$$f_{jt} - f_j^* = (1 - \eta \lambda_j)^t (f_{j0} - f_j^*) \tag{37}$$

637   where $f_j^*$ are the coefficients of the (mininum norm) interpolating solution.

# C   Complexity Bounds

## C.1   Rademacher Complexity

640   Given a family $\mathcal{G} \subset \mathbb{R}^{\mathcal{Z}}$ of real-valued functions on a probability space $(\mathcal{Z}, \rho)$, the empirical
641   Rademacher complexity of $\mathcal{G}$ with respect to a sample $\mathcal{S} = \{\mathbf{z}_1, \cdots \mathbf{z}_n\} \sim \rho^n$ is defined as [38]:

$$\widehat{\mathcal{R}}_{\mathcal{S}}(\mathcal{G}) = \mathbb{E}_{\boldsymbol{\sigma} \in \{\pm 1\}^n} \left[ \sup_{g \in \mathcal{G}} \frac{1}{n} \sum_{i=1}^n \sigma_i g(\mathbf{z}_i) \right], \tag{38}$$

642   where the expectation is over $n$ i.i.d uniform random variables $\sigma_1, \cdots \sigma_n \in \{\pm 1\}$. For any $n \geq 1$,
643   the Rademacher complexity with respect to samples of size $n$ is then $\mathcal{R}_n(\mathcal{G}) = \mathbb{E}_{\mathcal{S} \sim \rho^n} \widehat{\mathcal{R}}_{\mathcal{S}}(\mathcal{G})$.

## C.2 Generalization Bounds

Generalization bounds based on Rademacher complexity are standard [7, 38]. We give here one instance of such a bound, relevant for classification task.

**Setup.** We consider a family $\mathcal{F}$ of functions $f_{\mathbf{w}} \colon \mathcal{X} \to \mathbb{R}^c$ that output a score or probability $f_{\mathbf{w}}(\mathbf{x})[y]$ for each class $y \in \{1 \cdots c\}$ (we take $c = 1$ for binary classification). The task is to find a predictor $f_{\mathbf{w}} \in \mathcal{F}$ with small expected classification error, which can be expressed e.g. as

$$L_0(f_{\mathbf{w}}) = \mathbb{P}_{(\mathbf{x},y)\sim\rho} \{\mu(f_{\mathbf{w}}(\mathbf{x}), y) < 0\} \tag{39}$$

where $\mu(f(\mathbf{x}), y)$ denotes the **margin**,

$$\mu(f(\mathbf{x}), y) = \begin{cases} f(\mathbf{x})y & \text{binary case} \\ f(\mathbf{x})[y] - \max_{y' \neq y} f(\mathbf{x})[y'] & \text{multiclass case} \end{cases} \tag{40}$$

**Margin Bound.** We consider the **margin loss**,

$$\ell_\gamma(f_{\mathbf{w}}(\mathbf{x}), y)) = \phi_\gamma(\mu(f_{\mathbf{w}}(\mathbf{x}), y)) \tag{41}$$

where $\gamma > 0$, and $\phi_\gamma$ is the **ramp** function: $\phi_\gamma(u) = 1$ if $u \leq 0$, $\phi(u) = 0$ if $u > \gamma$ and $\phi(u) = 1 - u/\gamma$ otherwise. We have the following bound for the expected error (39). With probability at least $1 - \delta$ over the draw $\mathcal{S} = \{\mathbf{z}_i = (\mathbf{x}_i, y_i)\}_{i=1}^n$ of size $n$, the following holds for all $f_{\mathbf{w}} \in \mathcal{F}$ [38, Theorems 4.4.and 8.1]:

$$L_0(f_{\mathbf{w}}) \leq \widehat{L}_\gamma(f_{\mathbf{w}}) + 2\widehat{\mathcal{R}}_\mathcal{S}(\ell_\gamma(\mathcal{F}, \cdot)) + 3\sqrt{\frac{\log\frac{2}{\delta}}{2n}} \tag{42}$$

where $\widehat{L}_\gamma(f_{\mathbf{w}}) = \frac{1}{n}\sum_{i=1}^n \ell_\gamma(f_{\mathbf{w}}(\mathbf{x}_i), y_i)$ is the empirical margin error and $\ell_\gamma(\mathcal{F}, \cdot)$ is the **loss class**,

$$\ell_\gamma(\mathcal{F}, \cdot) = \{(\mathbf{x}, y) \mapsto \ell_\gamma(f_{\mathbf{w}}(\mathbf{x}), y) \,|\, f_{\mathbf{w}} \in \mathcal{F}\} \tag{43}$$

For binary classifiers, because $\phi_\gamma$ is $1/\gamma$-Lipschitz, we have in addition

$$\mathcal{R}_\mathcal{S}(\ell_\gamma(\mathcal{F}, \cdot)) \leq \frac{1}{\gamma}\mathcal{R}_\mathcal{S}(\mathcal{F}) \tag{44}$$

by Talagrand's contraction lemma [36] (see e.g. Mohri et al. [38, lemma 4.2] for a detailed proof).

## C.3 Complexity Bounds: Proofs

We first derive standard bounds for the linear families (70) of scalar functions ($c = 1$):

$$\mathcal{F}_{M_A}^A = \{f_{\mathbf{w}} \colon \mathbf{x} \mapsto \langle \mathbf{w}, \Phi(\mathbf{x})\rangle \,|\, \|\mathbf{w}\|_A \leq M_A\} \tag{45}$$

**Theorem 6.** *The empirical Rademacher complexity of $\mathcal{F}_{M_A}^A$ is bounded as,*

$$\widehat{\mathcal{R}}_\mathcal{S}(\mathcal{F}_{M_A}^A) \leq (M_A/n)\sqrt{\mathrm{Tr}\boldsymbol{K}_A} \tag{46}$$

*where $(\boldsymbol{K}_A)_{ij} = k_A(\mathbf{x}_i, \mathbf{x}_j)$ is the kernel matrix associated to the rescaled features $A^{-1}\Phi$.*

*Proof.* We use the notation of Section **??**. For given Rademacher variables $\boldsymbol{\sigma} \in \{\pm 1\}^n$, we have,

$$\begin{aligned} \sup_{f \in \mathcal{F}_{M_A}^A} \sum_{i=1}^n \sigma_i f(\mathbf{x}_i) &= \sup_{\|\mathbf{w}\|_A \leq M_A} \sum_{i=1}^n \sigma_i\langle \mathbf{w}, \Phi(\mathbf{x}_i)\rangle \\ &= \sup_{\|A^\top\mathbf{w}\|_2 \leq M_A} \sum_{i=1}^n \sigma_i\langle A^\top\mathbf{w}, A^{-1}\Phi(\mathbf{x}_i)\rangle \\ &= \sup_{\|\tilde{\mathbf{w}}\|_2 \leq M_A} \langle \tilde{\mathbf{w}}, \sum_{i=1}^n \sigma_i A^{-1}\Phi(\mathbf{x}_i)\rangle \\ &= M_A \left\|\sum_{i=1}^n \sigma_i A^{-1}\Phi(\mathbf{x}_i)\right\|_2 \\ &= M_A \sqrt{\boldsymbol{\sigma}^\top \boldsymbol{K}_A \boldsymbol{\sigma}} \end{aligned} \tag{47}$$

From (47) and the definition (38) we obtain:

$$
\begin{aligned}
\widehat{\mathcal{R}}_{\mathcal{S}}(\mathcal{F}_{M_A}^A) &= \frac{M_A}{n} \mathbb{E}_{\boldsymbol{\sigma}} \left[ \sqrt{\boldsymbol{\sigma}^{\top} \boldsymbol{K}_A \boldsymbol{\sigma}} \right] \\
&\leq \frac{M_A}{n} \sqrt{\mathbb{E}_{\boldsymbol{\sigma}} \left[ \boldsymbol{\sigma}^{\top} \boldsymbol{K}_A \boldsymbol{\sigma} \right]} \\
&\leq \frac{M_A}{n} \sqrt{\mathrm{Tr} \boldsymbol{K}_A}
\end{aligned}
\tag{48}
$$

where we used Jensen's inequality to pass $\mathbb{E}_{\sigma}$ under the root, and the properties that $\mathbb{E}[\sigma_i] = 0$ and $\sigma_i^2 = 1$ for all $i$. $\qquad\square$

We now extend the result to the families (16) of learning flows:

$$
\mathcal{F}_{\boldsymbol{m}}^{\boldsymbol{A}} = \{ f_{\mathbf{w}} \colon \mathbf{x} \mapsto \textstyle\sum_t \langle \delta \mathbf{w}_t, \Phi(\mathbf{x}) \rangle \mid \|\delta \mathbf{w}_t\|_{A_t} \leq m_t \}
\tag{49}
$$

**Theorem 7** (Theorem 3 restated)**.** *The empirical Rademacher complexity of $\mathcal{F}_{\boldsymbol{m}}^{\boldsymbol{A}}$ is bounded as,*

$$
\widehat{\mathcal{R}}_{\mathcal{S}}(\mathcal{F}_{\boldsymbol{m}}^{\boldsymbol{A}}) \leq \textstyle\sum_t (m_t/n) \sqrt{\mathrm{Tr} \boldsymbol{K}_{A_t}}
\tag{50}
$$

*where $(\boldsymbol{K}_{A_t})_{ij} = k_{A_t}(\mathbf{x}_i, \mathbf{x}_j)$ is the kernel matrix associated to the rescaled features $A_t^{-1}\Phi$.*

*Proof.* This is simple extension of the previous proof:

$$
\begin{aligned}
\sup_{f \in \mathcal{F}_{\boldsymbol{m}}^{\boldsymbol{A}}} \sum_{i=1}^n \sigma_i f(\mathbf{x}_i) &= \sup_{\|\delta \mathbf{w}_t\|_{A_t} \leq m_t} \sum_{i=1}^n \sigma_i \sum_t \langle \delta \mathbf{w}_t, \Phi(\mathbf{x}_i) \rangle \\
&= \sum_t \sup_{\|\tilde{\delta} \mathbf{w}_t\|_2 \leq m_t} \langle \tilde{\delta} \mathbf{w}_t, \sum_{i=1}^n \sigma_i A_t^{-1} \Phi(\mathbf{x}_i) \rangle \\
&= \sum_t m_t \sqrt{\boldsymbol{\sigma}^{\top} \boldsymbol{K}_{A_t} \boldsymbol{\sigma}}
\end{aligned}
\tag{51}
$$

and we conclude as in (48). $\qquad\square$

Finally, we note that the same result can be formulated in terms of an evolving feature map $\Phi_t = A_t^{-1}\Phi$ with kernel $k_t(\mathbf{x}, \tilde{\mathbf{x}}) = \langle \Phi_t(\mathbf{x}), \Phi_t(\tilde{\mathbf{x}}) \rangle$ In fact by reparametrization invariance, the function updates can also be written as $\delta f_{\mathbf{w}_t}(\mathbf{x}) = \langle \tilde{\delta} \mathbf{w}_t, \Phi_t(\mathbf{x}) \rangle$ where $\tilde{\delta} \mathbf{w}_t = A_t^{\top} \delta \mathbf{w}_t$. The function class (16) can equivalently be written as $\mathcal{F}_{\boldsymbol{m}}^{\boldsymbol{A}} = \mathcal{F}_{\boldsymbol{m}}^{\boldsymbol{\Phi}}$ where $\boldsymbol{\Phi}$ denotes a fixed sequence of feature maps, $\boldsymbol{\Phi} = \{\Phi_t\}_t$ and

$$
\mathcal{F}_{\boldsymbol{m}}^{\boldsymbol{\Phi}} = \{ f_{\mathbf{w}} \colon \mathbf{x} \mapsto \textstyle\sum_t \langle \tilde{\delta} \mathbf{w}_t, \Phi_t(\mathbf{x}) \rangle \mid \|\tilde{\delta} \mathbf{w}_t\|_2 \leq m_t \}
\tag{52}
$$

In this formulation, Theorem 3 becomes:

**Theorem 3bis.** *The empirical Rademacher complexity of $\mathcal{F}_{\boldsymbol{m}}^{\boldsymbol{\Phi}}$ is bounded as,*

$$
\widehat{\mathcal{R}}_{\mathcal{S}}(\mathcal{F}_{\boldsymbol{m}}^{\boldsymbol{\Phi}}) \leq \textstyle\sum_t (m_t/n) \sqrt{\mathrm{Tr} \boldsymbol{K}_t}
\tag{53}
$$

*where $(\boldsymbol{K}_t)_{ij} = k_t(\mathbf{x}_i, \tilde{\mathbf{x}}_j)$ is the kernel matrix associated to the feature map $\Phi_t$.*

### C.4 Bounds for Multiclass Classification

The generalization bound (42) is based on the **margin loss class** (43). In this section, we show how to bound $\widehat{\mathcal{R}}_{\mathcal{S}}(\ell_\gamma(\mathcal{F}, \cdot))$ in terms of tangent kernels for the original class $\mathcal{F}$ of functions $f_{\mathbf{w}} \colon \mathcal{X} \to \mathbb{R}^c$ instead. Although the proof is adapted from standard techniques, to our knowledge Lemma C.4 and Theorem 8 below are new results. In what follows, we denote by $\mu_{\mathcal{F}}$ the margin class,

$$
\mu_{\mathcal{F}} = \{ (\mathbf{x}, y) \to \mu(f_{\mathbf{w}}(\mathbf{x}), y) \mid f_{\mathbf{w}} \in \mathcal{F} \}
\tag{54}
$$

where $\mu(f_{\mathbf{w}}(\mathbf{x}), y))$ is the margin (40). We also define, for each $y \in \{1 \cdots c\}$,

$$
\mathcal{F}_y = \{ \mathbf{x} \mapsto f_{\mathbf{w}}(\mathbf{x})[y] \mid f_{\mathbf{w}} \in \mathcal{F} \}, \quad \mu_{\mathcal{F}, y} = \{ \mathbf{x} \mapsto \mu(f_{\mathbf{w}}(\mathbf{x}), y) \mid f_{\mathbf{w}} \in \mathcal{F} \}
\tag{55}
$$

The following inequality holds:

$$
\widehat{\mathcal{R}}_{\mathcal{S}}(\ell_\gamma(\mathcal{F}, \cdot)) \leq \frac{c}{\gamma} \sum_{y=1}^c \widehat{\mathcal{R}}_{\mathcal{S}}(\mathcal{F}_y)
\tag{56}
$$

*Proof.* We first follow the first steps of the proof of Mohri et al. [38, Theorem 8.1] to show that

$$\widehat{\mathcal{R}}_S(\ell_\gamma(\mathcal{F}, \cdot)) \leq \frac{1}{\gamma} \sum_{y=1}^c \widehat{\mathcal{R}}_S(\mu_{\mathcal{F},y}) \tag{57}$$

We reproduce these steps here for completeness: first, it follows from the $1/\gamma$-Lipschitzness of the ramp loss $\phi_\gamma$ in (41) and Talagrand's contraction lemma [38, lemma 4.2] that

$$\widehat{\mathcal{R}}_S(\ell_\gamma(\mathcal{F}, \cdot)) \leq \frac{1}{\gamma} \widehat{\mathcal{R}}_S(\mu_{\mathcal{F}}) \tag{58}$$

Next, we write

$$
\begin{aligned}
\widehat{\mathcal{R}}_S(\mu_{\mathcal{F}}) &:= \frac{1}{n} \mathbb{E}_{\boldsymbol{\sigma}} \left[ \sup_{f_{\mathbf{w}} \in \mathcal{F}} \sum_{i=1}^n \sigma_i \mu(f_{\mathbf{w}}(\mathbf{x}_i), y_i) \right] \\
&= \frac{1}{n} \mathbb{E}_{\boldsymbol{\sigma}} \left[ \sup_{f_{\mathbf{w}} \in \mathcal{F}} \sum_{i=1}^n \sigma_i \sum_{y=1}^c \mu(f_{\mathbf{w}}(\mathbf{x}_i), y) \, \delta_{y,y_i} \right] \\
&= \frac{1}{n} \sum_{y=1}^c \mathbb{E}_{\boldsymbol{\sigma}} \left[ \sup_{f_{\mathbf{w}} \in \mathcal{F}} \sum_{i=1}^n \sigma_i \mu(f_{\mathbf{w}}(\mathbf{x}_i), y) \, \delta_{y,y_i} \right] \tag{59}
\end{aligned}
$$

where $\delta_{y,y_i} = 1$ if $y = y_i$ and 0 otherwise; the second inequality follows from the sub-additivity of sup. Substituting $\delta_{y,y_i} = \frac{1}{2}(\epsilon_i + \frac{1}{2})$ where $\epsilon_i = 2\delta_{y,y_i} - 1 \in \{\pm 1\}$, we obtain

$$
\begin{aligned}
\widehat{\mathcal{R}}_S(\mu_{\mathcal{F}}) &\leq \frac{1}{2n} \sum_{y=1}^c \mathbb{E}_{\boldsymbol{\sigma}} \left[ \sup_{f_{\mathbf{w}} \in \mathcal{F}} \sum_{i=1}^n (\epsilon_i \sigma_i) \mu(f_{\mathbf{w}}(\mathbf{x}_i), y) \right] + \frac{1}{2n} \sum_{y=1}^c \mathbb{E}_{\boldsymbol{\sigma}} \left[ \sup_{f_{\mathbf{w}} \in \mathcal{F}} \sum_{i=1}^n \sigma_i \mu(f_{\mathbf{w}}(\mathbf{x}_i), y) \right] \\
&= \sum_{y=1}^c \frac{1}{n} \mathbb{E}_{\boldsymbol{\sigma}} \left[ \sup_{f_{\mathbf{w}} \in \mathcal{F}} \sum_{i=1}^n \sigma_i \mu(f_{\mathbf{w}}(\mathbf{x}_i), y) \right] \\
&= \sum_{y=1}^c \widehat{\mathcal{R}}_S(\mu_{\mathcal{F},y}) \tag{60}
\end{aligned}
$$

Together with (58), this leads to (57).

Now, spelling out $\mu(f_{\mathbf{w}}(\mathbf{x}_i, y))$ gives

$$
\begin{aligned}
\widehat{\mathcal{R}}_S(\mu_{\mathcal{F},y}) &= \frac{1}{n} \mathbb{E}_{\boldsymbol{\sigma}} \left[ \sup_{f_{\mathbf{w}} \in \mathcal{F}} \sum_{i=1}^n \sigma_i (f_{\mathbf{w}}(\mathbf{x}_i)[y] - \max_{y' \neq y} f_{\mathbf{w}}(\mathbf{x}_i)[y']) \right] \\
&= \widehat{\mathcal{R}}_S(\mathcal{F}_y) + \frac{1}{n} \mathbb{E}_{\boldsymbol{\sigma}} \left[ \sup_{f_{\mathbf{w}} \in \mathcal{F}} \sum_{i=1}^n (-\sigma_i) \max_{y' \neq y} f_{\mathbf{w}}(\mathbf{x}_i)[y'] \right] \\
&= \widehat{\mathcal{R}}_S(\mathcal{F}_y) + \frac{1}{n} \mathbb{E}_{\boldsymbol{\sigma}} \left[ \sup_{f_{\mathbf{w}} \in \mathcal{F}} \sum_{i=1}^n \sigma_i \max_{y' \neq y} f_{\mathbf{w}}(\mathbf{x}_i)[y'] \right] \\
&\leq \widehat{\mathcal{R}}_S(\mathcal{F}_y) + \widehat{\mathcal{R}}_S(\mathcal{G}_y) \tag{61}
\end{aligned}
$$

where $\mathcal{G}_y = \{\max\{f_{y'} : y' \neq y\} \mid f_{y'} \in \mathcal{F}_{y'}\}$. Now Mohri et al. [38, lemma 8.1] show that $\widehat{\mathcal{R}}_S(\mathcal{G}_y) \leq \sum_{y' \neq y} \widehat{\mathcal{R}}_S(\mathcal{F}_{y'})$. This leads to

$$
\begin{aligned}
\sum_{y=1}^c \widehat{\mathcal{R}}_S(\mu_{\mathcal{F},y}) &\leq \sum_{y=1}^c \widehat{\mathcal{R}}_S(\mathcal{F}_y) + \sum_{y=1}^c \sum_{\substack{y'=1 \\ y' \neq y}}^c \widehat{\mathcal{R}}_S(\mathcal{F}_{y'}) \\
&= \sum_{y=1}^c \widehat{\mathcal{R}}_S(\mathcal{F}_y) + (c-1) \sum_{y=1}^c \widehat{\mathcal{R}}_S(\mathcal{F}_y) \\
&= c \sum_{y=1}^c \widehat{\mathcal{R}}_S(\mathcal{F}_y) \tag{62}
\end{aligned}
$$

Substituting in (57) finishes the proof. $\qquad\square$

In the linear case, this results leads to analogous theorems as in C.3 in the multiclass setting. For example, considering the linear families of functions $\mathcal{X} \to \mathbb{R}^c$,

$$\mathcal{F}_{M_A}^A = \{\mathbf{x} \mapsto f_{\mathbf{w}}(\mathbf{x})[y] := \langle \mathbf{w}, \Phi(\mathbf{x})[y] \rangle \mid \|\mathbf{w}\|_A \leq M_A\} \tag{63}$$

where $(\mathbf{x}, y) \mapsto \Phi(\mathbf{x})[y]$ is some joint feature map, we have the following

**Theorem 8.** *The emp. Rademacher complexity of the margin loss class $\ell_\gamma(\mathcal{F}_{M_A}^A, \cdot)$ is bounded as,*

$$\widehat{\mathcal{R}}_{\mathcal{S}}(\ell_\gamma(\mathcal{F}_{M_A}^A, \cdot)) \leq (c^{3/2} M_A / \gamma n)\sqrt{\mathrm{Tr}\boldsymbol{K}_A} \tag{64}$$

*where $(\boldsymbol{K}_A)_{ij}^{yy'}$ is the kernel $nc \times nc$ matrix associated to the rescaled features $A^{-1}\Phi(\mathbf{x})[y]$.*

*Proof.* Eq.56, and Theorem 8 applied to each linear family $\mathcal{F}_y$ of (scalar) functions leads to

$$\widehat{\mathcal{R}}_{\mathcal{S}}(\ell_\gamma(\mathcal{F}_{M_A}^A, \cdot)) \leq \frac{c}{\gamma} \sum_{y=1}^c \frac{M_A}{n}\sqrt{\mathrm{Tr}\boldsymbol{K}_A^{yy}} \tag{65}$$

where $\mathrm{Tr}\boldsymbol{K}_A^{yy} := \sum_{i=1}^n (\boldsymbol{K}_A)_{ii}^{yy}$ is computed w.r.t to the indices $i = 1, ..., n$ for fixed $y$. Passing the average $\frac{1}{c}\sum_{y=1}^c$ under the root using Jensen inequality, we conclude:

$$
\begin{aligned}
\widehat{\mathcal{R}}_{\mathcal{S}}(\ell_\gamma(\mathcal{F}_{M_A}^A, \cdot)) &\leq \frac{c^2 M_A}{\gamma n}\sqrt{\frac{1}{c}\sum_{y=1}^c \mathrm{Tr}\boldsymbol{K}_A^{yy}} \\
&= \frac{c^{3/2} M_A}{\gamma n}\sqrt{\mathrm{Tr}\boldsymbol{K}_A}
\end{aligned} \tag{66}
$$

$\square$

## C.5 Linear models: Which Norm for Measuring Capacity?

We consider a family $\mathcal{F}$ of scalar functions $f_{\mathbf{w}}(\mathbf{x}) = \langle \mathbf{w}, \Phi(\mathbf{x}) \rangle$ linearly parametrized by $\mathbf{w} \in \mathbb{R}^P$, where $\Phi$ is a fixed mapping of the input space $\mathcal{X}$ into $\mathbb{R}^P$. Given a training set $\mathcal{S}$ of size $n$, we denote by $\boldsymbol{\Phi} = [\Phi(\mathbf{x}_1), \cdots \Phi(\mathbf{x}_n)]^\top$ the $n \times P$ feature matrix and by $\boldsymbol{y} = [y_1 \cdots y_n]^\top$ the label vector. We are interested in the 'overparametrized' regime: we assume $P \geq n$. We write the SVD of the feature matrix as $\boldsymbol{\Phi} = \sum_{j=1}^n \sqrt{\lambda_j}\boldsymbol{u}_j\boldsymbol{v}_j^\top$, where $\lambda_1 \geq \cdots \geq \lambda_n$ are ranked in nonincreasing order. We will consider the minimum $\ell^2$ norm interpolators [27],

$$\mathbf{w}^* = \boldsymbol{\Phi}^\top \boldsymbol{K}^{-1}\boldsymbol{y} = \sum_{j=1}^n \frac{\boldsymbol{u}_j^\top \boldsymbol{y}}{\sqrt{\lambda_j}}\boldsymbol{v}_j \tag{67}$$

A standard approach is to measure capacity in terms of the $\ell^2$ norm the weight vector. Considering

$$\mathcal{F}_M = \{f_{\mathbf{w}} : x \mapsto \langle \mathbf{w}, \Phi(x) \rangle \mid \|\mathbf{w}\|_2 \leq M\}, \tag{68}$$

the Rademacher complexity of $\mathcal{F}_M$ can be bounded as [6, Lemma 22]:

$$\widehat{\mathcal{R}}_{\mathcal{S}}(\mathcal{F}_M) \leq (M/n)\|\boldsymbol{\Phi}\|_\mathrm{F} \tag{69}$$

where $\|\boldsymbol{\Phi}\|_\mathrm{F}$ is the Froebenius norm of the feature matrix.[15]

Is the $\ell^2$ norm a good capacity measure, even for algorithms biased towards low $\ell^2$ norm solutions? If the distribution of solutions $\mathbf{w}_{\mathcal{S}}^*$, where $\mathcal{S} \sim \rho^n$, is reasonably isotropic, taking the smallest $\ell^2$ ball containing them (with high probability) gives an accurate description of the class of trained models. However for very anisotropic distributions, the solutions do not fill any such ball so describing trained models in terms of $\ell^2$ balls is wasteful [53]. For the minimum $\ell^2$ norm interpolators (67), the solution distribution typically inherits the anisotropy of the features. For example, if $y_i = \bar{y}(\mathbf{x}_i) + \varepsilon_i$ where $\varepsilon_i \sim \mathcal{N}(0, \sigma^2)$, the covariance of the solutions with respect to noise is $\mathrm{cov}_{\boldsymbol{\varepsilon}}[\mathbf{w}^*, \mathbf{w}^*] = \sum_j \frac{\sigma^2}{\lambda_j}\boldsymbol{v}_j\boldsymbol{v}_j^\top$, which scales as $1/\lambda_j$ along $\boldsymbol{v}_j$.

---

[15]Note that $\|\boldsymbol{\Phi}\|_\mathrm{F} = \sqrt{\mathrm{Tr}\boldsymbol{K}}$ where $\boldsymbol{K} = \boldsymbol{\Phi}\boldsymbol{\Phi}^\top$ is the kernel matrix.

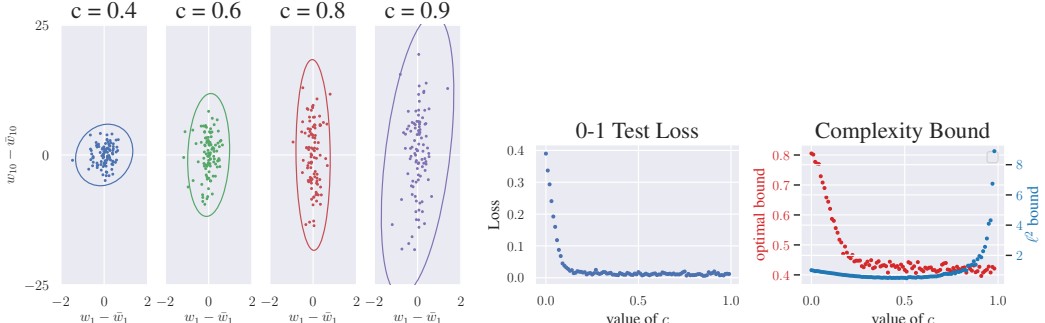

Figure 11: **Left:** 2D projection of the minimum-$\ell^2$-norm interpolators $\mathbf{w}_{\mathcal{S}}^*$, $\mathcal{S} \sim \rho^n$, for linear models $f_{\mathbf{w}} = \langle \mathbf{w}, \Phi_c \rangle$, as the feature scaling factor varies from 0 (white features) to 1 (original, anisotropic features). For larger $c$, the solutions scatter in a very anisotropic way. **Right:** Average test classification loss and complexity bounds (69) (blue plot) for the solution vectors $\mathbf{w}_{\mathcal{S}}^*$, as we increase the scaling factor $c$. As feature anisotropy increases, the bound becomes increasingly loose and fails to reflect the shape of the test error. By contrast, the bound (71) optimized as in Proposition 9 (red plot) does not suffer from this problem.

To visualize this on a simple setting, consider $P$ random Fourier features [49], fit on 1D data $\mathbf{x}$ modelled by $N$ equally spaced points in $[-a, a]$. In this setting, the (true) feature map is represented by a $N \times P$ matrix with SVD $\Phi = \sum_j \sqrt{l_j} \psi_j \varphi_j^\top$. The labels are given by $y(\mathbf{x}) = \text{sign}(\psi_1(\mathbf{x}))$. To highlight the effect of feature anisotropy, we further rescale the singular values as $l_j^c = 1 + c(l_j - 1)$ so as to interpolate between whitened features ($c=0$) and the original ones ($c=1$). We set $P=N=1000$. Fig 11 (left) shows 2D projections in the plane $(\varphi_1, \varphi_{10})$ of (centered) solutions $\mathbf{w}_{\mathcal{S}}^* - \mathbb{E}_{\mathcal{S}} \mathbf{w}_{\mathcal{S}}^*$, for a pool of 100 (sub)samples $\mathcal{S}$ of size $n = 50$, for increasing values of the scaling factor $c$. As $c$ approaches 1, the solutions begin to scatter in a very anisotropic way in parameter space; as shown in Fig 11 (right), the complexity bound (69) (blue plot) becomes increasingly loose and fails to reflect the shape of the test error.

We emphasize that this issue is about the choice of norm and not about complexity-based bounds *per se*. In fact, note that anisotropies can in principle be compensated by a suitable linear reparametrization $\mathbf{w} \mapsto A^\top \mathbf{w}$, $\Phi \mapsto A^{-1}\Phi$. Any such $A$ can be viewed as defining a new norm $\|\mathbf{w}\|_A := \sqrt{\mathbf{w}^\top g_A \mathbf{w}}$ induced by the metric $g_A = AA^\top$. The following classes

$$\mathcal{F}_{M_A}^A = \{f_{\mathbf{w}} : \mathbf{x} \mapsto \langle \mathbf{w}, \Phi(\mathbf{x}) \rangle \mid \|\mathbf{w}\|_A \leq M_A\}, \tag{70}$$

define a much richer set of complexity classes than (68), represented by ellipsoids of all shapes in parameter space. A direct extension of the standard result (69) yields:

$$\widehat{\mathcal{R}}_{\mathcal{S}}(\mathcal{F}_{M_A}^A) \leq (M_A/n)\|A^{-1}\mathbf{\Phi}^\top\|_{\text{F}} \tag{71}$$

in terms of the Froebenius norm of the rescaled feature matrix.[16] More meaningful norms than the $\ell^2$ norm can be found by optimizing the bound (71) with $M_A = \|\mathbf{w}^*\|_A$, over a given class of reparametrization matrices $A$. We give an example of this in the following Proposition.

**Proposition 9.** *Consider the class of reparametrization matrices* $A_\nu = \sum_{j=1}^n \sqrt{\nu_j} \boldsymbol{v}_j \boldsymbol{v}_j^\top + \mathbb{1}_{\text{span}\{\boldsymbol{v}_1, \cdots \boldsymbol{v}_n\}^\perp}$, *which act as mere rescaling* $\lambda_j \to \lambda_j/\nu_j$ *of the singular values of the feature matrix. Any minimizer of (71) for the mininum $\ell^2$-norm interpolator takes the form*

$$\nu_j^* = \kappa \frac{\sqrt{\lambda_j}}{|\boldsymbol{v}_j^\top \mathbf{w}^*|} = \kappa \frac{\lambda_j}{|\boldsymbol{u}_j^\top \boldsymbol{y}|} \tag{72}$$

*where $\kappa > 0$ is a constant independant of $j$.*

Note that in the context of Proposition 9, the optimal norm $\|\cdot\|_{A_{\nu^*}}$ depends both on the feature geometry – through the singular values – and on the task – through the labels –. As shown in Fig 1 (right, red plot), in the random Fourier feature setting, the corresponding bound has a much nicer behaviour than the standard bound (69) based on the $\ell^2$ norm.

---

[16] We also have $\|A^{-1}\mathbf{\Phi}^\top\|_{\text{F}} = \sqrt{\text{Tr}\boldsymbol{K}_A}$ where $\boldsymbol{K}_A = \mathbf{\Phi} g_A^{-1} \mathbf{\Phi}^\top$ is the rescaled kernel matrix.

## D  CKA and Spectral Entropy

We make explicit a couple of metrics used in Section 3.

**Centered kernel alignment (CKA).**  We used CKA [18] to measure the similarity between tangent features and labels. Given two kernel matrices $\boldsymbol{K}$ and $\boldsymbol{K}'$ in $\mathbb{R}^{r \times r}$, it is defined as

$$\text{CKA}(\boldsymbol{K}, \boldsymbol{K}') = \frac{\text{Tr}[\boldsymbol{K}_c \boldsymbol{K}'_c]}{\|\boldsymbol{K}_c\|_F \|\boldsymbol{K}'_c\|_F} \in [0, 1] \tag{73}$$

where the $c$ subscript denotes the feature centering operation, i.e. $\boldsymbol{K}_c = C\boldsymbol{K}C$ where $C = I_r - \frac{1}{r}\mathbf{1}\mathbf{1}^T$ is the centering matrix. The normalization by the Froebenius norm makes CKA invariant under isotropic rescaling.

Let $\boldsymbol{Y} \in \mathbb{R}^{nc}$ be the vector resulting from the concatenation of the one-hot label representations $\boldsymbol{Y}_i \in \mathbb{R}^c$ of the $n$ samples. Similarity with the labels is measured through CKA with the rank-one kernel $\boldsymbol{K_Y} := \boldsymbol{Y}\boldsymbol{Y}^\top$,

$$\text{CKA}(\boldsymbol{K}, \boldsymbol{K_Y}) = \frac{\boldsymbol{Y}^\top \boldsymbol{K}_c \boldsymbol{Y}}{\|\boldsymbol{K}_c\|_F \|\boldsymbol{K}_{\boldsymbol{Y}c}\|_F} \tag{74}$$

**Effective rank.**  We used a notion of effective rank based on **spectral entropy** [50]. Given a kernel matrix $\boldsymbol{K}$ with (strictly) positive eigenvalues $\lambda_1, \cdots, \lambda_n$, let

$$\mu_j = \lambda_j / \text{Tr}\boldsymbol{K}, \quad \text{Tr}\boldsymbol{K} = \sum_{i=1}^{n} \lambda_j \tag{75}$$

be the trace-normalized eigenvalues. The effective rank is defined as [50]:

$$\text{erank} = \exp(H(\mu_1, \cdots \mu_n)) \tag{76}$$

where $H(\mu)$ is the Shannon entropy given by

$$H(\mu_1, \cdots \mu_n) = -\sum_{j=1}^{n} \mu_j \log(\mu_j) \tag{77}$$

This effective rank is a real number between 1 and $n$, upper bounded by $\text{rank}(\boldsymbol{K})$, which measures the 'uniformity' of the spectrum through the entropy.

## E  Experiments: Details and Additions

### E.1  Synthetic Experiment of Fig 3

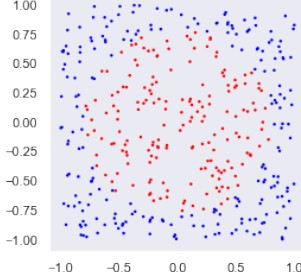 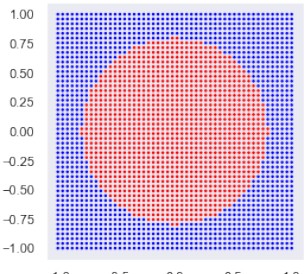

Figure 12: Disk dataset. **Left**: Training set of $n = 500$ points $(\mathbf{x}_i, y_i)$ where $\mathbf{x} \sim \text{Unif}[-1, 1]^2$, $y_i = 1$ if $\|x_i\|_2 \leq r = \sqrt{2/\pi}$ and $-1$ otherwise. **Right**: Large test sample (2500 points forming a $50 \times 50$ grid) used to evaluate the tangent kernel. In our experiment, we trained a 6-layer deep 256-unit wide MLP by full batch gradient descent of binary cross entropy.

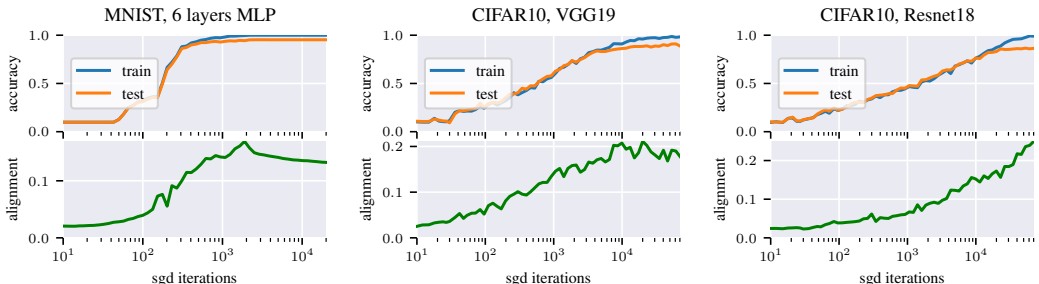

Figure 13: Evolution of the CKA between the tangent kernel and the class label kernel $K_Y = YY^T$ measured on a held-out test set for different architectures: **(left)** 6 layers of 80 hidden units MLP on MNIST **(middle)** VGG19 on CIFAR10 **(right)** Resnet18 on CIFAR10. We observe an increase of the alignment to the target function.

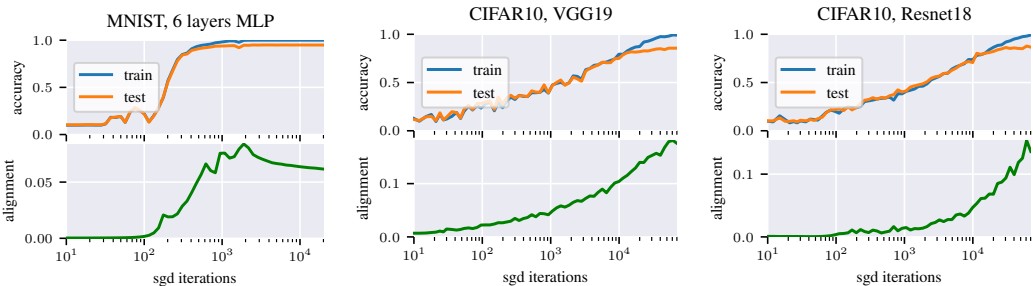

Figure 14: Same as figure 13 but without centering the kernel. Evolution of the uncentered kernel alignment between the tangent kernel and the class label kernel $K_Y = YY^T$ measured on a held-out test set for different architectures: **(left)** 6 layers of 80 hidden units MLP on MNIST **(middle)** VGG19 on CIFAR10 **(right)** Resnet18 on CIFAR10. We observe an increase of the alignment to the target function.

## E.2 More alignment plots

## E.3 More plots on spectra

## E.4 Ablation: Effect of depth on alignment

In order to study the influence of the architecture on the alignment effect, we measure the CKA for different networks and different initialization as we increase the depth. The results in Fig 16 suggest that the alignment effect is magnified as depth increases. We also observe that the ratio of the maximum alignment between easy and difficult examples is increased with depth, but stays high for a smaller number of iterations.

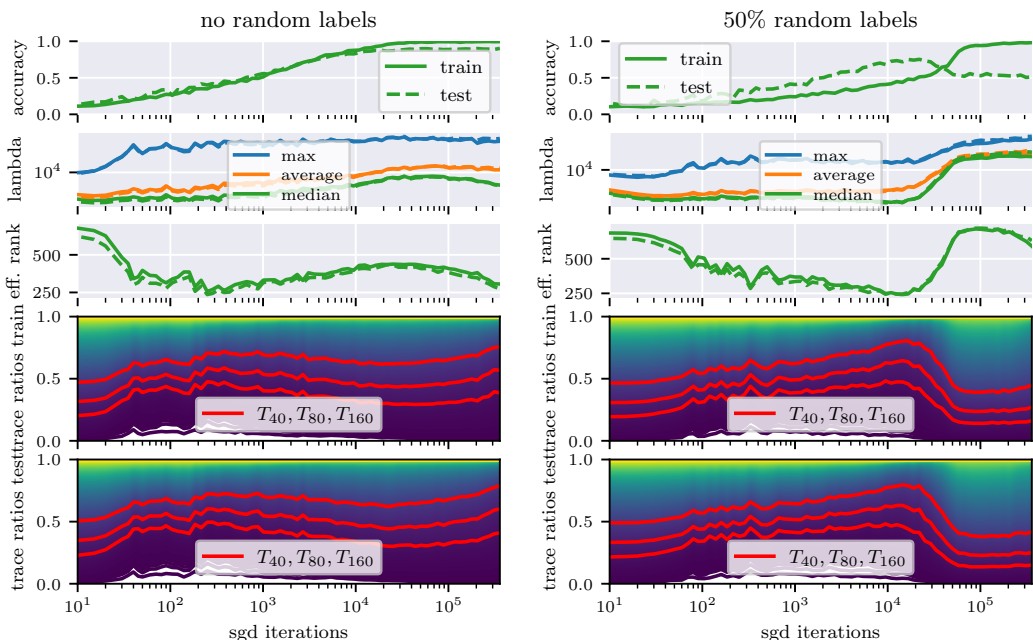

Figure 15: Evolution of tangent kernel spectrum, effective rank and trace ratios of a VGG19 trained by SGD with batch size 100, learning rate 0.003 and momentum 0.9 on dataset **(left)** CIFAR10 and **(right)** CIFAR10 with 50% random labels. We highlight the top 40, 80 and 160 trace ratios in **red**. The small effective rank of the kernel biases the training procedure towards a few top eigenvectors, as can also be observed by remarking that the trace ratio $T_{40}$ account for $\sim 50\%$ of the total trace.

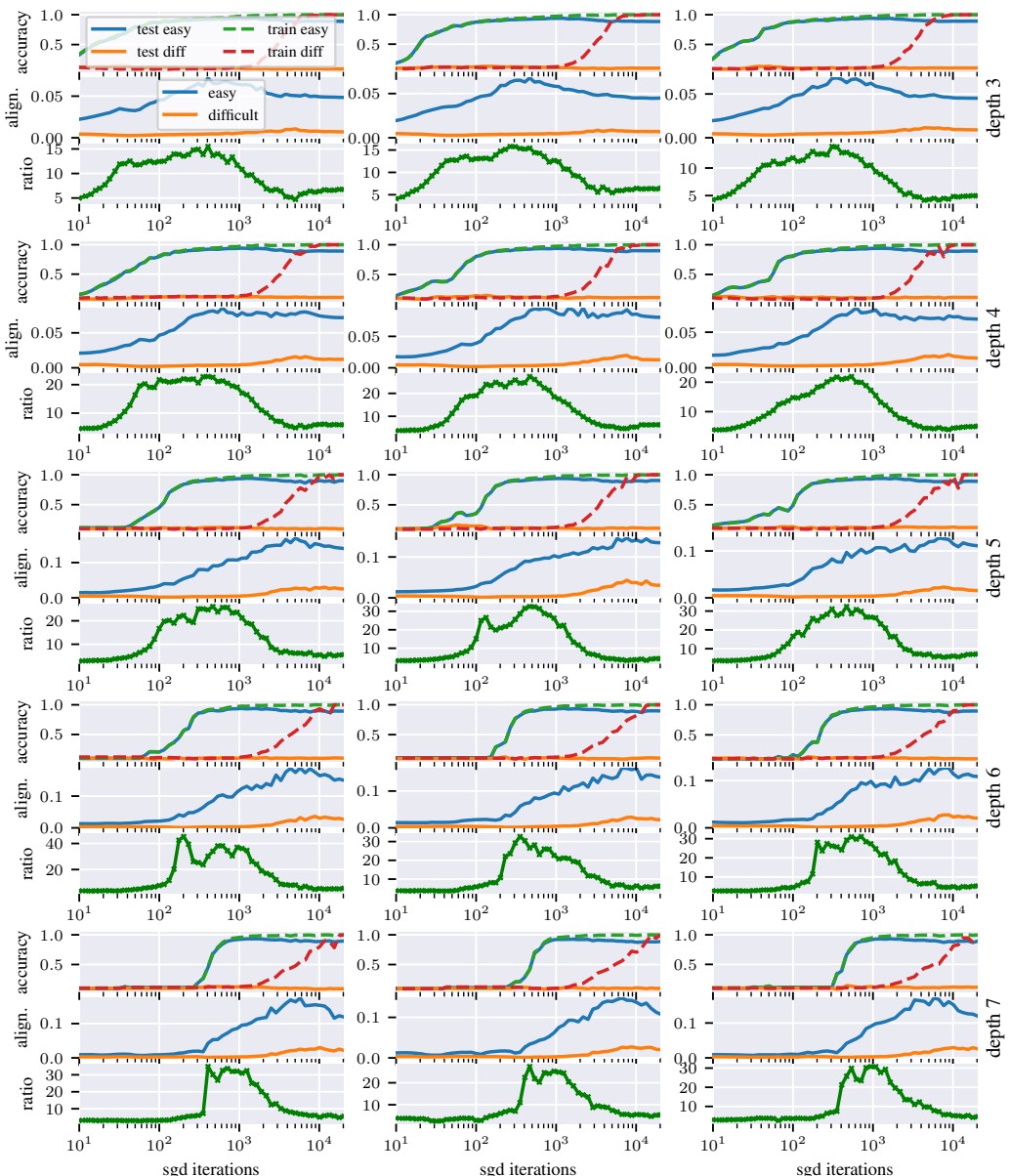

Figure 16: Effect of depth on alignment. 10.000 MNIST examples with 1000 random labels MNIST examples trained with learning rate=0.01, momentum=0.9 and batch size=100 for MLP with hidden layers size 60 and **(in rows)** varying depths **(in columns)** varying random initialization/minibatch sampling. As we increase the depth, the alignment starts increasing later in training and increases faster; and the ratio between easy and difficult alignments reaches a higher value.

