# OpenReview forum: "Implicit Regularization via Neural Feature Alignment"
_NeurIPS.cc/2020/Workshop/DL-IG — NeurIPSW 2020: DL-IG Poster_

### Official Review · AnonReviewer2 · 2020-10-26
**Review of "Implicit Regularization via Neural Feature Alignment"**

**Rating:** 7
**Confidence:** 2

**Review:**

This paper proposes an explanation for the generalization capability of neural networks using a geometric perspective. The authors show by multiple experiments that the neural tangent features are concentrated on a small number of eigenvectors of the tangent kernel basis. This observation leads the authors to the conclusion that during the training procedure, the neural network adapts to the intrinsic complexity of the problem and thus generalizes well despite having high capacity. The authors also propose a notion of effective rank to describe the intrinsic complexity and show that this quantity correlates with generalization.

The results in the paper are interesting and novel. I would have appreciated if the authors explained equation (2) in the preliminaries with more details. There is a lot of notation in equation (2) and it is somewhat central to paper.

---

### Official Review · AnonReviewer1 · 2020-11-07
**Review of "Implicit Regularization via Neural Feature Alignment"**

**Rating:** 9
**Confidence:** 4

**Review:**

The geometric point of view to explain generalizability of deep networks is indeed an interesting avenue to explore in this research area. The experimental evidences appear to support the authors claim that the neural tangent features are concentrated on a small number of eigenvectors of the tangent kernel basis which in turn helps the network adapt to the intrinsic complexity of the problem.

The proposal presented in this work is interesting and worth exploring further. I recommend to accept the paper for the workshop.

---

### Decision · Program_Chairs · 2020-11-07

Accept (Poster)

---

> ### Author Response · Authors · 2020-12-13
> **Link to Poster**
>
> https://www.dropbox.com/s/9znx9fn48cdglti/Complexity_Poster_IG.pdf?dl=0